# CoLD: Counterfactually-Guided Length Debiasing for Process Reward Models in Mathematical Reasoning

## Abstract

Process Reward Models (PRMs) play a central role in evaluating and guiding multi-step reasoning in large language models (LLMs), especially for mathematical problem solving. However, we identify a pervasive length bias in existing PRMs: a tendency to assign higher scores to more verbose reasoning steps, regardless of their semantic content or logical validity. This bias undermines the reliability of reward predictions and leads to overly verbose outputs during inference. To address this issue, we propose **CoLD** (**Co**unterfactually-Guided **L**ength **D**ebiasing), a unified framework that mitigates length bias based on counterfactual reasoning and causal graph analysis through three components: (1) an explicit length-penalty module, (2) a trainable bias estimator to capture spurious length-related signals, and (3) a joint training strategy that disentangles semantic correctness from superficial length features. Extensive experiments on MATH500 and GSM-Plus show that CoLD consistently reduces reward–length correlation, improves accuracy in step selection, and encourages more concise, logically valid reasoning. These results demonstrate the effectiveness and practicality of CoLD in improving the fidelity and robustness of PRMs.

## 1 Introduction

Large language models (LLMs) have shown strong mathematical reasoning ability (OpenAI, 2023; Dubey et al., 2024; Zhu et al., 2024b; Shao et al., 2024; Liu et al., 2024a; Yang et al., 2025), yet their solutions often contain hidden reasoning errors despite yielding correct final answers (Lightman et al., 2023). Process Reward Models (PRMs) (Lightman et al., 2023; Wang et al., 2024b) were introduced to evaluate the logical soundness of intermediate steps and are now widely used in both error diagnosis and inference-time scaling strategies (Wu et al., 2024; Snell et al., 2024; Zhao et al., 2025). Recent works (Zhu et al., 2025; Wang et al., 2024a; o1 Team, 2024; Zou et al., 2025) typically formulate PRM training as a binary classification task, using either human-annotated or automatically generated data. Despite the im-

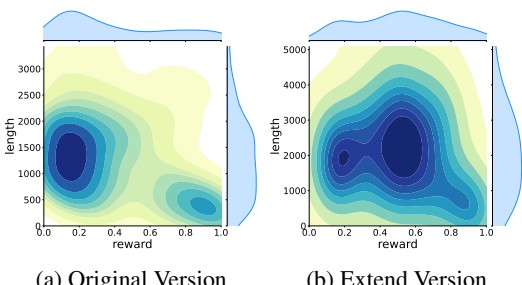

(a) Original Version        (b) Extend Version

Figure 1: The joint distribution of reward score (x-axis) and step length (y-axis). (a) Distribution for the original reasoning steps. (b) Distribution for the same steps after being paraphrased by DeepSeek-V3 for increased textual length and verbosity, while maintaining semantic and logical equivalence.

pressive performance of recent PRMs, we observe that their reward predictions can be unduly influenced by superficial features of reasoning steps, particularly their step-level textual length. We term this phenomenon **length bias**: *the tendency of PRMs to assign higher scores to textually longer or more verbose steps, even if their semantic content and logical correctness are identical to more concise counterparts.* Such bias undermines the reliability of PRMs, as it confuses surface-level verbosity with the quality of substantive reasoning.

To rigorously examine the length bias phenomenon, we design a controlled experiment that isolates step length while holding reasoning quality constant. Based on existing datasets (Li & Li, 2024), we

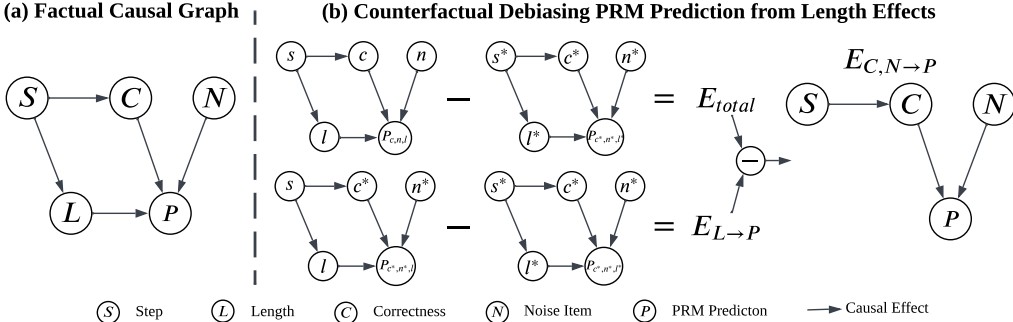

Figure 2: Causal Modeling to Eliminate Length Bias in Prediction with Causal Graph of Factors Affecting PRM Reward. Uppercase letters (e.g., $L$) represent random variables, lowercase letters (e.g., $l$) denote specific values, and lowercase letters with an asterisk (e.g., $l^*$) indicate fixed values. $E$ denotes the causal effect.

construct a semi-synthetic dataset and generate extended variants of each step by either duplicating the original step or prompting LLM to rewrite it with greater verbosity while preserving semantics and logical correctness. We then input both the original and extended versions into the same PRM, recording the reward scores and token lengths. As shown in Figure 1, the textually longer steps consistently receive higher rewards despite being logically equivalent to the originals. This suggests that PRMs exploit step length as a spurious shortcut, inflating scores in ways that undermine their reliability in assessing true reasoning quality.

To further analyze the factors influencing PRM predictions, we construct a causal graph (Pearl, 2009) that captures the qualitative relationships among key variables, as illustrated in Figure 2(a). The graph includes five nodes: the input step ($S$), its length ($L$), logical correctness ($C$), latent noise factors ($N$), and the PRM prediction ($P$).

This analysis reveals a central concern: PRM predictions are not solely determined by correctness. In the causal graph, the step $S$ determines both its length and correctness ($S \rightarrow L$, $S \rightarrow C$). Ideally, predictions should follow the path $S \rightarrow C \rightarrow P$, ensuring that scores reflect reasoning quality. However, we identify a spurious path $S \rightarrow L \rightarrow P$, indicating that verbosity directly affects the predicted reward—even when it adds no logical value. While latent noise factors such as fluency or problem type ($N$) may also impact $P$, their influence appears secondary. Additional analysis is provided in the Preliminary section 2.2.

To address this issue, we adopt a counterfactual perspective, framing length bias as the undesired change in a PRM's output that occurs when the step length is altered, while semantics and correctness remain fixed. This perspective motivates the goal of isolating and eliminating the score component attributable solely to length, as shown in Figure 2(b). Guided by this insight, we propose Counterfactually-Guided Length Debiasing (CoLD)—a unified framework consisting of three methods: (1) Length Penalty: A simple adjustment that subtracts a length-proportional term from the original PRM score, explicitly discouraging verbosity. (2) Bias Estimator: An auxiliary model trained to estimate the length-induced bias and subtract it from the PRM score, aiming to preserve correctness while restoring length invariance. (3) Joint Training: A unified training scheme that jointly optimizes the PRM and the Bias Estimator. By introducing input perturbations and regularization, the model is encouraged to disentangle semantic content from superficial length features.

Together, these components form a principled debiasing strategy. By explicitly modeling, estimating, and removing spurious length effects, CoLD reduces the correlation between reward and verbosity while maintaining semantic fidelity. Empirically, it improves selection accuracy and promotes concise, logically sound responses.

The main contributions of our paper are summarized as follows:

- We are the first to identify and empirically validate length bias in PRMs, showing the tendency of PRMs to assign higher scores to textually longer or more verbose steps, even if their semantic content and logical correctness are identical to more concise counterparts. Through causal graph analysis, we further reveal that step length is a confounding factor that distorts reward estimation and undermines the reliability of PRMs.

- We introduce the CoLD framework, grounded in counterfactual reasoning, to mitigate length bias in PRMs. CoLD significantly improves the fairness and accuracy of reward assessments.

- We augment public datasets with semi-synthetic examples and conduct extensive experiments on both original and augmented data. The results consistently demonstrate that our framework effectively mitigates length bias, enhancing the overall reliability and effectiveness of PRMs.

## 2 PRELIMINARY

### 2.1 FORMULATION

We consider the problem of assigning reward scores to intermediate reasoning steps in mathematical problem solving. Let $q$ denote a math question and $\mathbf{s} = \{s^1, s^2, \ldots, s^n\}$ be a sequence of solution steps. For each prefix $x^j = (q, s^{\leq j})$ comprising the first $j$ steps, a Process Reward Model (PRM) assigns a scalar reward:

$$r(x^j) \in (0, 1), \tag{1}$$

which is intended to reflect the semantic correctness of the partial solution up to step $j$.

To implement $r(x^j)$ using large language models (LLMs), we follow a scoring protocol that computes the reward based on the model's classification between correctness and incorrectness. Specifically, given $x^j$, we extract the logits of two special answer tokens (e.g., '+' for correct and '-' for incorrect), and apply a softmax to compute a score:

$$r(x^j) = \frac{\exp(l_+)}{\exp(l_+) + \exp(l_-)}, \tag{2}$$

where $l_+$ and $l_-$ denote the logits assigned to the positive and negative options, respectively.

### 2.2 CAUSAL-DRIVEN ANALYSIS

Causal graphs (Pearl, 2009), formally known as directed acyclic graphs, represent variables as nodes and encode causal relationships through directed edges. They offer a principled framework for modeling complex dependencies and reasoning about cause-and-effect mechanisms within a system. In this section, we leverage a causal perspective to analyze the reward prediction behavior of PRMs and draw insights that inform the design of our framework.

The previous Figure 2(a) illustrates the causal graph underlying PRM reward prediction, each node represents a causal factor, and each directed edge $A \to B$ indicates that $A$ exerts a direct causal influence on $B$.

- The causal graph consists of five nodes: $S, L, C, N$ and $P$. $S$ denotes the given problem-solving step, $L$ represents the length of the given step, $C$ represents the logical correctness of the problem-solving step. $N$ represents latent noise factors such as linguistic fluency or problem category. $P$ represents PRM's reward prediction for this step.

- For a given step, it determines both length and correctness, resulting in causal edges $S \to L$ and $S \to C$. While $L$ and $C$ have no direct causal link, a confounding relation may exist. For example, extremely short steps of only one or two words often fail to convey valid reasoning, reducing the chance of correctness.

- We then examine the key pathways through which $S$ influences $P$. Ideally, $P$ should depend only on correctness $C$, i.e., $S \to C \to P$, making PRM a reliable evaluator that assigns high rewards to logically valid steps.

- In practice, however, the alternative path $S \to L \to P$ also plays a substantial role. Step length $L$ positively influences $P$: more verbose steps tend to receive higher scores, despite conveying the same reasoning content as their shorter counterparts. This reveals a spurious shortcut exploited by the PRM, where verbosity is mistakenly rewarded.

- Finally, while unobserved noise $N$ may also directly influence $P$, as also shown in previous work Shen et al. (2023), its effect is comparatively minor.

Building on the causal analysis, it is crucial to distinguish the genuine causal effect of correctness $(S \to C \to P)$ from the spurious influence of length $(S \to L \to P)$. As only the former reflects true reasoning quality, PRM predictions should eliminate length effects during inference. This motivates a counterfactual objective: isolating the PRM score that remains invariant to length while holding

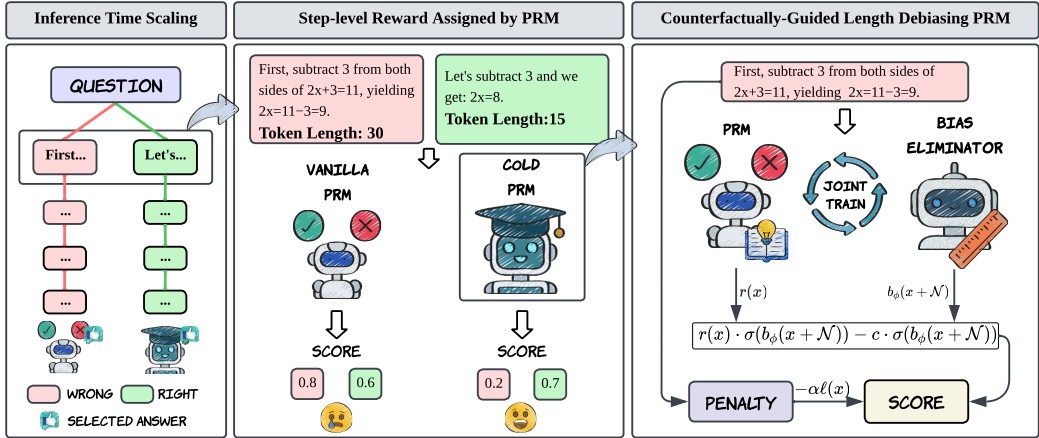

Figure 3: Overview of the Counterfactually-Guided Length Debiasing (CoLD) framework for Process Reward Models, in comparison to the vanilla PRMs.

correctness fixed, as illustrated in Figure 2(b). A more detailed theoretical analysis is presented in Appendix A. Based on the above analysis, we then introduce a debiasing framework designed to suppress superficial cues and preserve the integrity of step-level evaluation.

## 3 METHODOLOGY

In this section, we introduce our proposed Counterfactually-Guided Length Debiasing (**CoLD**) framework for Process Reward Models.

### 3.1 OVERVIEW OF COLD PRM

To address length bias in PRMs, we propose the CoLD framework. As illustrated in Figure 3, PRMs are used at inference time to guide the selection of high-quality reasoning paths, particularly in settings like best-of-N sampling, where multiple candidate solutions are generated and scored. In such scenarios, the effectiveness of the PRM has a direct impact on the final selection. However, vanilla PRMs tend to assign higher scores to more verbose reasoning steps, even when those steps are semantically incorrect. This leads to spurious preferences for textually longer but lower-quality responses over shorter, more accurate ones. To mitigate this issue, CoLD PRM systematically removes the reward bias caused by step length, effectively isolating it from the true evaluation of step quality.

Rather than relying on a single correction strategy, the CoLD PRM framework integrates three complementary methods to mitigate length bias: (1) an explicit penalty term that directly discourages verbosity by penalizing textually longer steps; (2) a learnable bias estimator that captures the reward component attributable to length; (3) a joint optimization scheme that synchronizes the training of the PRM and the bias estimator to enable consistent debiasing.

Under this framework, the final debiased reward $r^*(x)$ is computed as:

$$r^*(x) = r_\theta(x) \cdot \sigma(b_\phi(x + \mathcal{N})) - c \cdot \sigma(b_\phi(x + \mathcal{N})) - \alpha\ell(x), \quad (3)$$

where $r_\theta(x)$ denotes the score assigned by the PRM, $b_\phi(x + \mathcal{N})$ denotes the output of the bias estimator given the noise-injected input $x + \mathcal{N}$ ($\mathcal{N}$ denotes the Gaussian noise following Shen et al. (2023)), $\sigma(\cdot)$ is the sigmoid function, $c$ is a hyperparameter controlling the strength of bias correction, and $\alpha\ell(x)$ represents the length penalty term.

By integrating these components, CoLD PRM progressively refines reward estimation, disentangling semantic correctness from superficial features like length. This structured approach facilitates counterfactual debiasing in a principled manner. Further analytical insights into this formulation are provided in the following discussion.

## 3.2 LENGTH PENALTY

Length Penalty item $\alpha\ell(x)$ is a simple yet effective heuristic approach that explicitly penalizes verbosity by subtracting a length-proportional term from the original PRM score.

Here, $\alpha > 0$ is a hyperparameter controlling the penalty strength, and $\ell(x)$ denotes the token-level length of the reasoning step $x$. This formulation introduces an explicit dependency on length into the reward computation, encouraging concise responses and discouraging unnecessary verbosity.

## 3.3 BIAS ESTIMATOR

To more flexibly approximate the counterfactual difference attributable to step length, we introduce a learnable module $b_\phi(x)$, referred to as the Bias Estimator. The primary goal of the Bias Estimator is to capture and eliminate the spurious reward component that arises solely from variations in step length, thereby isolating the superficial length bias from the true semantic reward.

Instead of directly feeding the original input into the estimator, we apply noise perturbation to encourage the model to focus on non-semantic features. Specifically, the input is defined as $\hat{x} = x + \mathcal{N}$, where $\mathcal{N}$ denotes injected Gaussian noise (Shen et al., 2023). This design discourages $b_\phi$ from relying on semantic content and guides it to concentrate on surface-level factors such as length.

## 3.4 JOINT TRAINING OF PRM WITH BIAS ESTIMATOR

To mitigate the impact of length bias on PRM predictions, we employ a joint training strategy that simultaneously optimizes the Process Reward Model (PRM) $r_\theta(x)$ and the Bias Estimator $b_\phi(x)$. Ensuring a proper disentanglement between correctness and stylistic features, such as length, is achieved through the use of complementary correlation constraints.

We enforce complementary correlation constraints to ensure effective disentanglement between semantic correctness and stylistic factors such as length. Specifically, we measure the Pearson correlation between each module's output and the step length $\ell(x)$:

$$\rho_r = \frac{\mathrm{Cov}(r_\theta(x), \ell)}{\sigma_r \sigma_\ell} = \frac{\mathbb{E}\left[(r_\theta(x) - \mathbb{E}[r_\theta(x)])(\ell - \mathbb{E}[\ell])\right]}{\sigma_r \sigma_\ell}, \tag{4}$$

$$\rho_b = \frac{\mathrm{Cov}(b_\phi(x), \ell)}{\sigma_b \sigma_\ell} = \frac{\mathbb{E}\left[(b_\phi(x) - \mathbb{E}[b_\phi(x)])(\ell - \mathbb{E}[\ell])\right]}{\sigma_b \sigma_\ell}, \tag{5}$$

where $\mathrm{Cov}(\cdot, \cdot)$ denotes empirical covariance, and $\sigma_r$, $\sigma_b$, and $\sigma_\ell$ are the standard deviations. A small $\rho_r$ indicates that the PRM has reduced its reliance on length, while a large $\rho_b$ confirms that the Bias Estimator has effectively captured the length-dependent component. The corresponding correlation losses are defined as:

$$\mathcal{L}_{\mathrm{PRM}}(\theta) = \lambda_r \cdot \rho_r^2, \tag{6}$$

$$\mathcal{L}_{\mathrm{Bias}}(\phi) = -\lambda_b \cdot \rho_b^2, \tag{7}$$

where $\lambda_r$ and $\lambda_b$ control the strength of regularization. This asymmetric formulation promotes semantic fidelity in the PRM, while allowing the Bias Estimator to isolate and model spurious stylistic factors such as length.

However, relying solely on correlation-based constraints may lead to degenerate solutions that overlook semantic correctness. To ensure the debiased reward can still distinguish correct from incorrect steps, we add a cross-entropy loss as an additional supervision signal.

The composed reward is defined as $\hat{r}(x) = r_\theta(x) \cdot \sigma(b_\phi(\hat{x}))$, and is trained using correctness labels $y \in \{0, 1\}$ through a cross-entropy objective:

$$\mathcal{L}_{\mathrm{CE}}(\theta, \phi) = -\mathbb{E}_{(x,y) \sim \mathcal{D}}\Big[y \log \sigma(\hat{r}(x)) + (1 - y) \log(1 - \sigma(\hat{r}(x)))\Big], \tag{8}$$

where $\mathcal{D}$ denotes the training data distribution. This loss encourages the composed reward to align with the ground-truth correctness signal.

The final training objective combines all components:

$$\mathcal{L}_{\mathrm{Final}}(\theta, \phi) = \mathcal{L}_{\mathrm{CE}}(\theta, \phi) + \mathcal{L}_{\mathrm{PRM}}(\theta) + \mathcal{L}_{\mathrm{Bias}}(\phi). \tag{9}$$

Further explanation of the algorithm is provided in Appendix B.

Table 1: We compare our proposed CoLD method with existing RL debiasing approaches, with main results measured by best-of-16 accuracy. The best result is given in **bold**, and the second-best value is underlined.

| Policy Model | Debias Method | MATH500 | | GSM-Plus | | Avg | |
|---|---|---|---|---|---|---|---|
| | | ArithACC(%) | Length | ArithACC(%) | Length | ArithACC(%) | Length |
| Llama-3-70B-Instruct | Vanilla-Base-PRM | 44.8 | 555.7 | 71.0 | 324.6 | 57.9 | 440.1 |
| | + Length Penalty | 44.8 | 495.1 | 71.5 | 300.1 | 58.2 | 397.6 |
| | + Loose Lips Sink Ships | 45.8 | 435.7 | 72.5 | 302.5 | 59.2 | 368.9 |
| | + Adaptive Length Bias Mitigation | 43.0 | 594.8 | 72.2 | 364.3 | 57.6 | 479.6 |
| | + Uniform Average | 40.4 | 608.5 | 70.4 | 324.6 | 55.4 | 466.6 |
| | + Locally Weighted Regression | 39.0 | 590.7 | 69.6 | 299.9 | 47.2 | 445.3 |
| | + CoLD(w/o Joint) (Ours) | 48.0 | 370.4 | 72.1 | 258.6 | 60.1 | 314.5 |
| | + CoLD (Ours) | **49.2** | **313.2** | **73.8** | **202.5** | **61.5** | **257.9** |
| MetaMath-Mistral-7B | Vanilla-Base-PRM | 37.0 | 555.2 | 59.5 | 335.5 | 48.2 | 445.4 |
| | + Length Penalty | 37.6 | 448.8 | 59.1 | 313.9 | 48.4 | 381.4 |
| | + Loose Lips Sink Ships | 34.5 | 441.2 | 58.0 | 340.0 | 46.3 | 390.6 |
| | + Adaptive Length Bias Mitigation | 31.0 | 610.0 | 56.5 | 385.0 | 43.8 | 497.5 |
| | + Uniform Average | 33.2 | 609.9 | 56.3 | 363.8 | 44.8 | 486.9 |
| | + Locally Weighted Regression | 29.6 | 507.3 | 52.7 | 354.4 | 41.2 | 430.9 |
| | + CoLD(w/o Joint) (Ours) | **38.6** | 353.4 | 59.8 | 262.7 | 49.2 | 308.1 |
| | + CoLD (Ours) | 37.2 | **376.3** | **61.4** | **238.6** | **49.3** | **307.5** |
| Muggle-Math-13B | Vanilla-Base-PRM | 30.4 | 411.1 | 59.1 | 287.9 | 44.8 | 349.5 |
| | + Length Penalty | 30.0 | 376.1 | 59.3 | 280.5 | 44.7 | 328.3 |
| | + Loose Lips Sink Ships | 28.6 | 375.7 | 59.1 | 263.9 | 43.9 | 319.8 |
| | + Adaptive Length Bias Mitigation | 25.4 | 472.0 | 56.3 | 364.5 | 40.9 | 418.3 |
| | + Uniform Average | 27.4 | 440.8 | 55.3 | 328.7 | 41.4 | 384.8 |
| | + Locally Weighted Regression | 23.5 | 470.5 | 50.7 | 312.3 | 37.1 | 391.4 |
| | + CoLD(w/o Joint) (Ours) | 31.0 | 329.0 | 59.9 | **238.3** | 45.5 | 283.7 |
| | + CoLD (Ours) | **31.4** | **309.2** | **60.3** | 243.3 | **45.9** | **276.3** |

**Alternative Strategy.**  In addition to joint training, an alternative strategy is to train the Bias Estimator independently while keeping the PRM fixed. This approach offers greater flexibility and modularity, especially when modifying existing PRMs without re-training the whole model.

## 4 EXPERIMENTS

In this section, we present the experimental settings and results. The implementation code is available in the anonymous repository[1].

### 4.1 EXPERIMENT SETUP

**Datasets and Metrics**   We utilize human-annotated and large-scale automatically labeled datasets for training, and adopt established evaluation protocols to assess model performance. Specifically, we train our Models on two publicly available datasets PRM800K (Lightman et al., 2023) and Math-Shepherd (Wang et al., 2024b).

For evaluation, we adopt the BON@n metric (Lightman et al., 2023; Wang et al., 2024b), which quantifies the PRM's verification ability in selecting the most correct reasoning trajectory from a set of $n$ candidates. For each question, the PRM assigns scores to individual steps, and the overall score of a trajectory is determined by the minimum score among its constituent steps, following prior work (Wang et al., 2024b). Details about datasets can be found in Appendix C.1.

**Baselines and Implementation Details**   We consider a range of baselines and base models. Additional details, including further introductions to the baselines and base models, as well as hyperparameters and training sizes, are provided in Appendix C.1.

### 4.2 OVERALL PERFORMANCE

We present the performance of CoLD PRM, along with various baselines and base models, across two evaluation datasets: MATH500 and GSM-Plus. The main results are provided in Tables 1 and 2, and the key findings are outlined as follows:

---

[1]https://anonymous.4open.science/r/CoLD-PRM-CC68-ICLR2026/

Table 2: We apply our CoLD method across various base models, with primary results assessed using best-of-16 accuracy. The symbol ↑ indicates an increase in accuracy or length, while the symbol ↓ denotes a decrease in accuracy or length.

| Policy Model | PRM Model | MATH500 | | GSM-Plus | | Avg | |
|---|---|---|---|---|---|---|---|
| | | ArithACC(%) | Length | ArithACC(%) | Length | ArithACC(%) | Length |
| Llama-3-70B-Instruct | Vanilla-Math-Shepherd-PRM | 46.4 | 458.9 | 72.3 | 268.7 | 59.4 | 363.8 |
| | CoLD(w/o Joint)-Math-Shepherd-PRM | 47.2↑ | 332.4↓ | 72.5↑ | 258.0↓ | 59.85↑ | 295.2↓ |
| | Vanilla-Base-PRM | 44.8 | 555.7 | 71.0 | 324.6 | 57.9 | 440.1 |
| | CoLD(w/o Joint)-Base-PRM | 48.0↑ | 370.4↓ | 72.1↑ | 258.6↓ | 60.1↑ | 314.5↓ |
| | Vanilla-Qwen2.5-Math-PRM | 45.0 | 595.2 | 73.9 | 284.3 | 59.5 | 439.8 |
| | CoLD(w/o Joint)-Qwen2.5-Math-PRM | 49.0↑ | 330.6↓ | 74.4↑ | 217.6↓ | 61.7↑ | 274.1↓ |
| | Vanilla-EurusPRM-Stage1 | 49.0 | 546.6 | 73.5 | 308.2 | 61.3 | 427.4 |
| | CoLD(w/o Joint)-EurusPRM-Stage1 | 50.6↑ | 282.1↓ | 73.8↑ | 237.7↓ | 62.2↑ | 259.9↓ |
| MetaMath-Mistral-7B | Vanilla-Math-Shepherd-PRM | 32.6 | 381.3 | 59.9 | 291.8 | 46.3 | 336.6 |
| | CoLD(w/o Joint)-Math-Shepherd-PRM | 32.6 | 300.1↓ | 60.4↑ | 279.9↓ | 46.5↑ | 290.0↓ |
| | Vanilla-Base-PRM | 37.0 | 555.2 | 59.5 | 335.5 | 48.2 | 445.4 |
| | CoLD(w/o Joint)-Base-PRM | 38.6↑ | 353.4 | 59.8↑ | 262.7↓ | 49.2↑ | 308.1↓ |
| | Vanilla-Qwen2.5-Math-PRM | 36.4 | 648.8 | 62.8 | 292.3 | 49.6 | 470.6 |
| | CoLD(w/o Joint)-Qwen2.5-Math-PRM | 39.4↑ | 338.4↓ | 62.5↑ | 235.0↓ | 51.0↑ | 286.7↓ |
| | Vanilla-EurusPRM-Stage1 | 43.2 | 324.0 | 65.4 | 323.6 | 54.3 | 357.7 |
| | CoLD(w/o Joint)-EurusPRM-Stage1 | 42.0↓ | 239.7↓ | 64.8↓ | 260.8↓ | 53.4↓ | 250.3↓ |
| Muggle-Math-13B | Vanilla-Math-Shepherd-PRM | 28.6 | 332.1 | 59.2 | 279.7 | 43.9 | 305.9 |
| | CoLD(w/o Joint)-Math-Shepherd-PRM | 28.8↑ | 262.7↓ | 59.1↑ | 263.3↓ | 44.0↑ | 263.0↓ |
| | Vanilla-Base-PRM | 30.4 | 411.1 | 59.1 | 287.9 | 44.8 | 349.5 |
| | CoLD(w/o Joint)-Base-PRM | 31.0↑ | 329.0↓ | 59.9↑ | 238.3↓ | 45.5↑ | 283.7↓ |
| | Vanilla-Qwen2.5-Math-PRM | 30.2 | 399.0 | 60.8 | 251.1 | 45.5 | 325.1 |
| | CoLD(w/o Joint)-Qwen2.5-Math-PRM | 31.8↑ | 297.1↓ | 60.3↓ | 222.9↓ | 46.1↑ | 260.0↓ |
| | Vanilla-EurusPRM-Stage1 | 34.0 | 359.1 | 57.6 | 300.6 | 45.8 | 328.4 |
| | CoLD(w/o Joint)-EurusPRM-Stage1 | 34.6↑ | 282.7↓ | 57.9↑ | 265.7↓ | 46.3↑ | 274.2↓ |

- CoLD PRM not only achieves state-of-the-art accuracy but also favors significantly shorter reasoning steps. This demonstrates that our method effectively mitigates the length bias in reward modeling—correct steps are no longer over-rewarded simply because they are longer.

- The effect is particularly pronounced on the MATH500 dataset, where our debiasing method yields both higher accuracy and more substantial reductions in solution length. This is likely because trajectories in MATH500 are more complex and verbose, making it more susceptible to length bias. By correcting for this bias, CoLD PRM is better able to select concise yet correct solutions in challenging scenarios.

- As shown in Table 1, while these methods effectively tackle bias issues in RL, they do not perform particularly well when applied to the length bias problem in PRMs, exhibiting subpar results in both accuracy and the length of the selected responses. In contrast, our CoLD framework demonstrates a marked improvement in both accuracy and response length optimization.

- Table 2 demonstrates that even without joint training, combining a bias estimator with length-penalty correction substantially improves performance across various base models. This highlights the effectiveness of these components and suggests that our framework can be flexibly applied to existing, pre-trained PRMs. In settings with limited computational resources, this provides a practical and efficient way to improve reward quality without retraining the entire model.

- It is worth noting that shorter length does not inherently equate to higher accuracy. For instance, Math-Shepherd-PRM selects shorter reasoning steps yet exhibits lower accuracy. A model that aggressively favors brevity may overlook essential reasoning steps. The goal of CoLD is not to shorten responses indiscriminately, but rather to reduce unnecessary verbosity while preserving or even improving accuracy.

## 4.3 ABLATION STUDY

To understand the contribution of each component in our CoLD PRM framework, we conduct a comprehensive ablation study by selectively removing the Joint Training, Bias Estimator, and Length Penalty variants. The performance of these variants is presented in Table 3, from which we can draw the following observations:

- Removing Joint Training leads to a noticeable increase in the average solution length across both datasets, indicating that without joint optimization, the reward model tends to favor textually longer reasoning trajectories. This suggests that joint training helps align the reward signal with downstream policy behavior, implicitly regularizing verbosity.

Table 3: We assess the performance of CoLD PRM variants using Best-of-16 search with solutions sampled from the Llama-3-70B-Instruct model. Component-wise ablations are conducted to evaluate the contribution of each module. Additional ablation results can be found in Table 5 and Table 6.

| Components | | | MATH500 | | GSM-Plus | | Avg | |
|---|---|---|---|---|---|---|---|---|
| Joint Train | Bias Estimator | Penalty | ArithACC(%) | Length | ArithACC(%) | Length | ArithACC(%) | Length |
| ✓ | ✓ | ✓ | **49.2** | **313.2** | **73.8** | **202.5** | **61.5** | **257.9** |
| ✓ | ✓ | ✗ | 48.2 | 494.9 | 73.2 | 237.4 | 60.7 | 366.2 |
| ✗ | ✓ | ✓ | 48.0 | 370.4 | 72.1 | 258.6 | 60.1 | 314.5 |
| ✗ | ✓ | ✗ | 44.8 | 525.0 | 71.5 | 303.6 | 58.2 | 414.3 |
| ✗ | ✗ | ✓ | 44.8 | 495.1 | 71.5 | 300.1 | 58.2 | 397.6 |
| ✗ | ✗ | ✗ | 44.8 | 555.7 | 71.0 | 324.6 | 57.9 | 440.2 |

- Excluding the Bias Estimator similarly results in inflated solution lengths, while the accuracy remains relatively stable. This points to the Bias Estimator's role in explicitly identifying and correcting for length-related confounding effects in the reward scores, encouraging the model to recognize correctness independent of surface-level length cues.

- Using the Length Penalty alone reduces length but fails to improve correctness, and can even lead to a degradation in overall accuracy. This suggests that a naive penalty on length, when not informed by bias estimation, may remove beneficial reasoning steps along with redundant ones, highlighting the risk of applying heuristic penalties without semantic guidance.

## 4.4 DEBIAS VISUALIZATION

To better understand the effect of our debiasing framework, we visualize the reward distributions after applying CoLD. As discussed in the introduction and shown in Figure 1, the reward model favors longer reasoning steps even when they are semantically equivalent to shorter ones.

After applying CoLD, we observe a marked improvement in the alignment between the rewards assigned to original and extended steps. As shown in Figure 4, the distributions of scores become substantially more consistent, indicating that the model no longer favors more verbose steps. This suggests that CoLD PRM successfully removes the spurious correlation between step length and reward.

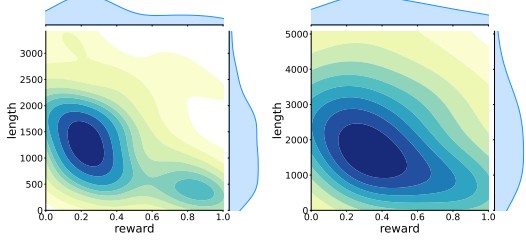

(a) Original Version     (b) Extend Version

Figure 4: The joint distribution of rewards and step lengths after debiasing on both original and length-augmented steps.

Notably, the debiased model places greater emphasis on the semantic correctness and logical validity of each step, rather than its superficial length. This demonstrates that our method not only improves quantitative performance but also corrects length bias in a principled manner.

## 4.5 SCALING WIDTH STUDY

In this section, we analyze the impact of our CoLD method under varying numbers of best-of-N samples. Experiments were conducted with $N = 2, 4, 8, 16$.

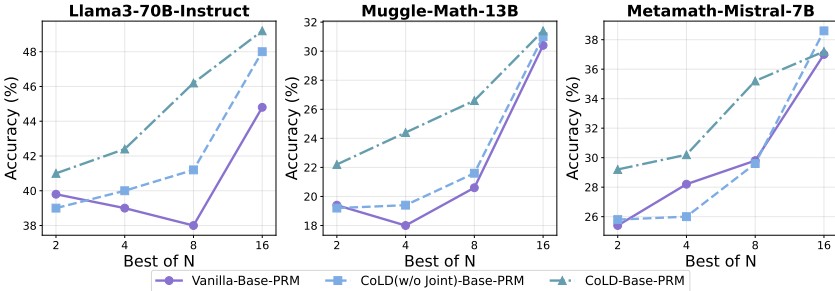

Figure 5: Performance of CoLD under different numbers of best-of-N samples on the Math dataset across different policy models.

Table 4: Performance of PRMs as reward signals in policy optimization.

| Reward Signal Source | Accuracy(%) | Length |
|---|---|---|
| Vanilla ReasonFlux-PRM | 54.4 | 600.9 |
| CoLD ReasonFlux-PRM | 57.6 | 543.2 |

As shown in Figure 5, our CoLD PRM consistently achieves a clear advantage across different scaling widths. For the CoLD PRM without joint training, while exhibiting slightly lower accuracy than the vanilla PRM at a few points, it outperforms the vanilla counterpart for most values of $n$. For the vanilla PRM, however, we observe that in some cases increasing the scaling width can paradoxically lead to a decline in accuracy despite the expectation that a larger $n$ should provide more candidate solutions and thus improve performance. This counterintuitive result highlights the substantial impact of length bias on the model's ability to correctly assess solution steps.

### 4.6 DOWNSTREAM RL APPLICATION

To more comprehensively demonstrate CoLD's effectiveness on downstream tasks, we apply CoLD in a reinforcement learning setting. Specifically, our experiments are conducted within the ReasonFlux-PRM framework (Zou et al., 2025). We use Qwen2.5-Math-1.5B-Instruct as the policy model and adopt GRPO (Group Relative Policy Optimization) (Shao et al., 2024) as the RL optimization algorithm. CoLD is integrated into the ReasonFlux-PRM and compared against its vanilla counterpart. For evaluation, we use MATH500 (Lightman et al., 2023) as the test benchmark.

As shown in the Table 4, incorporating CoLD not only improves the accuracy of the policy model in answering questions, but also reduces the output length. This indicates that CoLD effectively mitigates the length bias exhibited by the PRM, providing the model with more precise and meaningful reward signals during downstream reinforcement learning training.

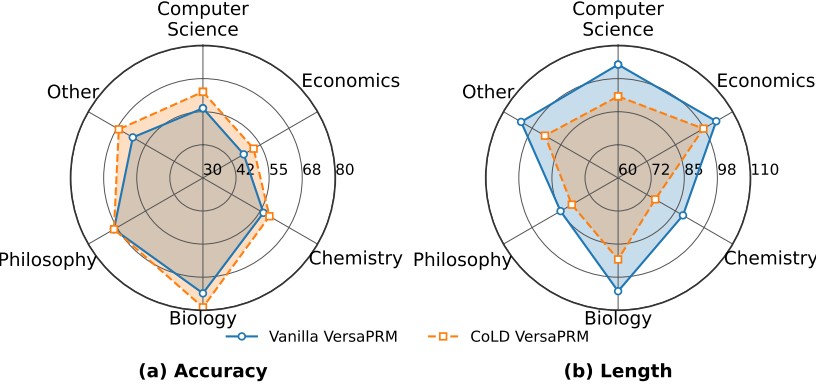

Figure 6: Performance of CoLD under different numbers of best-of-N samples on the Math dataset across different policy models.

### 4.7 CROSS-DOMAIN GENERALIZATION STUDY

To evaluate whether CoLD can generalize to domains beyond its original scope, we integrate it into VersaPRM (Zeng et al., 2025), a process reward model designed to operate across heterogeneous domains. We assess its performance on the MMLU-Pro (Wang et al., 2024c) benchmark, a robust and challenging massive multi-task understanding dataset created to rigorously test the capabilities of large language models. CoLD is applied within VersaPRM across several representative domains, including Computer Science, Economics, Chemistry, Biology, Philosophy, and Other.

Based on the results shown in the Figure 6, when applying CoLD on top of VersaPRM, we observe consistent improvements in both accuracy and the reduction of response length across these heterogeneous domains. This demonstrates that the debiasing principle behind CoLD does not rely on domain-specific structures found in mathematical reasoning, but instead addresses a general phenomenon of superficial-feature bias in step-level reward modeling.

### 4.8 ADDITIONAL EXPERIMENTS

**Hyperparameter Study** We conduct a systematic analysis of the relevant hyperparameters. Due to space limitations, the specific details can be found in the Appendix E.

**Other Benchmark** We conducted additional experiments using ProcessBench (Zheng et al., 2024), a mainstream evaluation benchmark for PRMs. Due to space limitations, detailed information is available in the Appendix F.

**Evaluation Using Stronger Policy Model** We further include Qwen3-Next-80B-A3B-Instruct Yang et al. (2025) and DeepSeek-v3.1 Liu et al. (2024a) in our evaluation in Appendix G.

**Bias Estimator Interpretability** We conduct interpretability analyses focusing on its sensitivity to input length versus semantic content in Appendix H.

## 5 RELATED WORKS

### 5.1 PROCESS REWARD MODELS

Process Reward Models (PRMs) enable fine-grained step-level supervision for model reasoning, addressing traditional Outcome Reward Models (ORMs) (Cobbe et al., 2021) limitation of only scoring final outputs. This design mitigates "spurious correctness" and boosts stability in complex tasks. A key challenge for PRM training is high annotation costs: PRM800K (Lightman et al., 2023) was the first human-annotated dataset, while later works (Wang et al., 2023; Luo et al., 2024; Zhang et al., 2025b) used LLMs and Monte Carlo (MC) methods to reduce costs. PRMs have advanced for both RL optimization (Rizvi et al., 2025; Chen et al., 2025; Wang et al., 2024a; Cheng et al., 2025; Zhang et al., 2025a; Feng et al., 2025) and inference time scaling (Ma et al., 2023; Xie et al., 2025; Zhao et al., 2025; Setlur et al., 2024; Hu et al., 2025b;a). Despite these advances, few PRM-related works have focused on the length bias issue inherent in PRMs—and this is precisely the problem addressed by our proposed CoLD PRM.

### 5.2 CAUSAL DEBIAS

Causal debias has emerged as a fundamental strategy across diverse machine learning domains (Zhu et al., 2024a; Yu et al., 2025; Sun et al., 2025). For example, Zhang et al. (2024) mitigates bias in multi-hop fact verification by applying front-door adjustment to reasoning paths and estimating causal effects via random walks. Chisca et al. (2024) reduce LLM bias by leveraging causal paths to design prompts that emphasize factual knowledge over stereotypes. Zhou et al. (2023) propose a fine-tuning framework that integrates causal intervention and invariant risk minimization to suppress reliance on non-causal, bias-inducing factors. Liu et al. (2025) introduces a counterfactual-enhanced framework that debiases multimodal sentiment classification through adaptive contrastive learning. Liu et al. (2024b) propose a causal framework with data augmentation to filter out irrelevant artifacts. Zhan et al. (2023) presents a counterfactual training strategy for Med-VQA, removing spurious linguistic correlations via causal intervention. Farzam & Sapiro analyzes how differential privacy can bias causal estimates and proposes robust regression techniques to correct such distortions. While these methods effectively address domain-specific confounders, none directly tackle the unique length bias (Park et al., 2024; Dubois et al., 2024) in process reward models (PRMs). Our proposed CoLD PRM fills this gap by disentangling semantic correctness from spurious correlations with output length.

## 6 CONCLUSION

This paper addresses the critical problem of length bias in Process Reward Models (PRMs). We introduce CoLD, a Counterfactually-Guided Length Debiasing framework, to tackle this issue. Specifically, CoLD incorporates a length penalty, a bias estimator, and joint training to mitigate the undue influence of response length on reward scores. Extensive experiments demonstrate that CoLD effectively mitigates length-related bias, leading to PRMs that are both more accurate and robust. Extensive experiments on real-world datasets demonstrate that CoLD PRM effectively removes length-related bias, highlighting the robustness and effectiveness of our approach.

ETHICS STATEMENT

This work studies length bias in Process Reward Models using publicly available mathematical reasoning datasets (MATH500 and GSM-Plus), without involving human subjects, sensitive data, or privacy concerns. Our proposed method, CoLD, aims to improve fairness and robustness in evaluating reasoning steps by mitigating spurious correlations, thereby reducing potential risks of biased or unreliable predictions. We affirm that this research complies with the ICLR Code of Ethics, and all authors have read and acknowledged the Code in full.

REPRODUCIBILITY STATEMENT

We have taken extensive steps to ensure the reproducibility of our work. The CoLD framework is detailed in Sections 3 and 4, while hyperparameters, datasets, and implementation details are provided in Appendix C. Randomness in all experiments is controlled through the setting of seeds. Our training is based on two publicly available datasets, PRM800K (Lightman et al., 2023) and Math-Shepherd (Wang et al., 2024b), and evaluation is conducted on datasets curated by Li & Li (2024), which draw from MATH500 (Hendrycks et al., 2024) and GSM-Plus (Li et al., 2024). To further support reproducibility, we provide anonymous source code in https://anonymous.4open.science/r/CoLD-PRM-CC68-ICLR2026/.

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

## A CAUSAL COUNTERFACTUAL ANALYSIS

In the causal graph shown in Figure 2, variables influence one another. For example, both correctness $C$ and step length $L$ affect the prediction $P$. Therefore, the value of $P$ can be determined by its ancestor nodes, mathematically expressed as:

$$P_{c,n,l} = P(C = c, N = n, L = \ell). \tag{10}$$

where $P(\cdot)$ denotes the value function associated with $P$.

To rigorously quantify the effect of these variables on the model prediction, we adopt counterfactual methods. The core idea of counterfactual analysis is to evaluate the outcome under hypothetical scenarios by selectively altering certain variables. For example, we can set $L$ to a fixed reference value $\ell^*$, which represents a counterfactual intervention that removes its actual effect from the system.

Table 5: We assess the performance of CoLD PRM variants using Best-of-16 search with solutions sampled from the Muggle-Math-13B model. Component-wise ablations are conducted to evaluate the contribution of each module.

| Components | | | MATH500 | | GSM-Plus | | Avg | |
|---|---|---|---|---|---|---|---|---|
| Joint Train | Bias Estimator | Penalty | ArithACC(%) | Length | ArithACC(%) | Length | ArithACC(%) | Length |
| ✓ | ✓ | ✓ | **31.4** | **309.2** | 60.3 | 243.3 | **45.9** | **276.3** |
| ✓ | ✓ | × | 30.6 | 387.7 | **60.4** | 273.1 | 45.5 | 330.4 |
| × | ✓ | ✓ | 31.0 | 329.0 | 59.9 | 238.3 | 45.5 | 283.7 |
| × | ✓ | × | 29.8 | 398.3 | 59.1 | 286.5 | 44.5 | 342.4 |
| × | × | ✓ | 30.0 | 376.1 | 59.3 | 280.5 | 44.7 | 328.3 |
| × | × | × | 30.4 | 411.1 | 59.1 | 287.9 | 44.8 | 349.5 |

Since $\ell^*$ is held constant, it acts as a reference condition with a consistent influence on downstream variables. Likewise, we counterfactually set $C$ to $c^*$ to isolate the combined effect of $C$ and $L$ on $P$. Under this setting, the total effect on $P$ can be formulated as:

$$E_{total} = P_{c,n,\ell} - P_{c^*,n^*,\ell^*} \tag{11}$$

Here, $C$ is also replaced by $c^*$ because $c^*$ also has an influence on $P$. $c^*$ represents the counterfactual value of $P$.

Moreover, based on the structure of the causal graph, we can decompose this total effect into two distinct components: (1) the effect of the correctness variable on the PRM prediction, i.e., the path $C \to P$, and (2) the effect of length, i.e., $L \to P$. To isolate the effect of length, we hold $C$ at its counterfactual value $c^*$ while allowing $L$ to vary, which gives:

$$E_{L \to P} = P_{c^*,n^*,\ell} - P_{c^*,n^*,\ell^*} \tag{12}$$

where $P_{c^*,L}$ represents the prediction when the observed length $L$ is retained, while the correctness signal $C$ has been counterfactually removed as shown in Figure 2(b).

We perform this counterfactual removal of $C$ because we aim to measure only the contribution of $L$ to the prediction, independent of $C$. This process cannot be directly computed from observational data and thus relies on causal reasoning, making it a canonical example of counterfactual causal inference.

Once $E_{total}$ and $E_{L \to P}$ are calculated, $E_{C,N \to P}$ can be obtained by subtracting the former from the latter, as illustrated in Figure 2:

$$
\begin{aligned}
E_{C,N \to P} &= E_{total} - E_{L \to P} \\
&= P_{c,n,\ell} - P_{c^*,n^*,\ell^*} - (P_{c^*,n^*,\ell} - P_{c^*,n^*,\ell^*}) \\
&= P_{c,n,\ell} - P_{c^*,n^*,l} \\
&= P(C = c, N = n, L = \ell) - P(C = c^*, N = n^*, L = \ell)
\end{aligned} \tag{13}
$$

This expression captures the isolated effect of the correctness signal $C$ on the model prediction $P$, with length $L$ held constant. As discussed in Section 3, the value $P(C = c, L = \ell)$ can be estimated using the composed reward $r_\theta(x)\sigma(b_\phi(x + \mathcal{N}))$, where $r_\theta(x)$ is the PRM score and $b_\phi(x + \mathcal{N})$ is the bias estimator output. On the other hand, $P(C = c^*, L = \ell)$ can be approximated by both $\sigma(b_\phi(x + \mathcal{N}))$ and $\alpha\ell(x)$.

During training, we only have access to supervision from the biased labels $P(C = c, L = \ell)$, so we use the composed reward $r_\theta(x)\sigma(b_\phi(x + \mathcal{N}))$ as the training target. However, during inference, we apply the debiasing approach described in Section 3.1 to compute the final corrected reward $r^*(x)$, which corresponds to the estimated value of $E_{C,N \to P}$ and reflects the true contribution of correctness, disentangled from length bias.

## B  ALGORITHM DESCRIPTION

Algorithm 1 summarizes the joint training procedure for the Process Reward Model (PRM) and the Bias Estimator. The training aims to disentangle semantic correctness from length-related bias in the reward predictions.

---

**Algorithm 1** Joint Training of PRM and Bias Estimator

---

**Require:** Training dataset $\mathcal{D} = \{(x_i, y_i)\}_{i=1}^N$, learning rates $\eta_r$, $\eta_b$, hyperparameters $\lambda_r$, $\lambda_b$, bias correction factor $c$
1: Initialize PRM $r_\theta(\cdot)$ and Bias Estimator $b_\phi(\cdot)$
2: **while** not converged **do**
3:      Sample mini-batch $\{(x_i, y_i)\}_{i=1}^B \sim \mathcal{D}$
4:      **for** each $x_i$ in batch **do**
5:          Inject noise: $\tilde{x}_i = x_i + \mathcal{N}$
6:          Compute $\hat{r}_i = r_\theta(x_i) \cdot \sigma\big(b_\phi(\tilde{x}_i)\big)$
7:      **end for**
8:      Compute cross-entropy loss:

$$\mathcal{L}_{\text{CE}} = -\frac{1}{B} \sum_{i=1}^B \left[ y_i \log \sigma(\hat{r}_i) + (1 - y_i) \log(1 - \sigma(\hat{r}_i)) \right]$$

9:      Compute Pearson correlations $\rho_r = \text{Corr}(r_\theta(x), \ell(x))$, $\rho_b = \text{Corr}(b_\phi(x), \ell(x))$
10:     Compute module-specific losses:

$$\mathcal{L}_{\text{PRM}} = \lambda_r \cdot \rho_r^2, \quad \mathcal{L}_{\text{Bias}} = -\lambda_b \cdot \rho_b^2$$

11:     Compute final losses:

$$\mathcal{L}_{\text{Final}} = \mathcal{L}_{\text{CE}} + \mathcal{L}_{\text{PRM}} + \mathcal{L}_{\text{Bias}}$$

12:     Update PRM parameters $\theta$: $\theta \leftarrow \theta - \eta_r \nabla_\theta(\mathcal{L}_{\text{CE}} + \mathcal{L}_{\text{PRM}})$
13:     Update Bias Estimator parameters $\phi$: $\phi \leftarrow \phi - \eta_b \nabla_\phi(\mathcal{L}_{\text{CE}} - \mathcal{L}_{\text{Bias}})$
14: **end while**

---

Table 6: We assess the performance of CoLD PRM variants using Best-of-16 search with solutions sampled from the MetaMath-Mistral-7B model. Component-wise ablations are conducted to evaluate the contribution of each module.

| Components | | | MATH500 | | GSM-Plus | | Avg | |
|---|---|---|---|---|---|---|---|---|
| Joint Train | Bias Estimator | Penalty | ArithACC(%) | Length | ArithACC(%) | Length | ArithACC(%) | Length |
| ✓ | ✓ | ✓ | 37.2 | 376.3 | **61.4** | **238.6** | **49.3** | **307.5** |
| ✓ | ✓ | ✗ | 36.2 | 552.6 | 61.3 | 289.7 | 48.8 | 421.2 |
| ✗ | ✓ | ✓ | **38.6** | 353.4 | 59.8 | 262.7 | 49.2 | 308.1 |
| ✗ | ✓ | ✗ | 38.0 | 554.1 | 58.9 | 322.8 | 48.5 | 438.4 |
| ✗ | ✗ | ✓ | 37.6 | 448.8 | 59.1 | 313.9 | 48.4 | 381.4 |
| ✗ | ✗ | ✗ | 30.4 | 411.1 | 59.4 | 335.5 | 44.9 | 373.3 |

At each iteration, a mini-batch of training samples is drawn from the dataset. For each sample, noise is injected into the input features before feeding them to the Bias Estimator to prevent it from capturing semantic information. The composed reward $\hat{r}(x)$ is then computed as the element-wise product of the PRM output and the sigmoid-activated Bias Estimator output.

The binary cross-entropy loss supervises the composed reward to align with ground-truth correctness labels. To encourage the PRM to focus on correctness rather than spurious length correlations, a regularization term penalizing the squared Pearson correlation between PRM predictions and step length is added. Conversely, the Bias Estimator is encouraged to maximize its correlation with step length to model the bias effectively.

Separate gradient updates are applied to the PRM and Bias Estimator parameters using their respective loss functions, allowing each module to specialize in modeling semantic correctness and length bias, respectively.

At inference time, the debiased reward is computed by correcting the PRM output with the estimated bias from the Bias Estimator, scaled by a hyperparameter controlling the bias removal magnitude.

## C EXPERIMENT DETAILS

### C.1 IMPLEMENTATION DETAILS

**Train Dataset**  We use both PRM800K and Math-Shepherd as training datasets. The first, PRM800K (Lightman et al., 2023), consists of 800,000 step-level correctness labels derived from the MATH dataset via extensive human annotation, offering a high-fidelity but annotation-expensive training resource. The second, Math-Shepherd (Wang et al., 2024b), comprises 400,000 automatically generated labels across both MATH and GSM8K problems. It provides scalable supervision without human involvement, allowing for cost-effective training at scale.

**Evaluation Dataset**  The evaluation datasets employed in our study are curated by Li & Li (2024), drawing from the MATH500 (Hendrycks et al., 2024) and GSM-Plus (Li et al., 2024) datasets. The collected trajectories are sampled from three mathematical solvers of varying model scales: MetaMath-Mistral-7B (Yu et al., 2023), MuggleMath-13B (Li et al., 2023), and Llama3-70B-Instruct (AI@Meta, 2024). In addition, we construct a semi-synthetic extension of this dataset to facilitate controlled evaluation of length-related biases. Specifically, for each original solution trajectory, we generate semantically equivalent but longer variants using two strategies: (1) duplicating individual steps to create trivially lengthened versions, and (2) prompting DeepSeek (Liu et al., 2024a) to rewrite each step with increased verbosity while preserving its semantic meaning and logical validity. These two transformation methods reflect two common patterns in model-generated outputs—verbatim repetition and verbose paraphrasing with limited substantive contribution. An example illustrating the two expansion methods can be found in Figure 7.

Based on the above, we construct a test set consisting of 16 solutions per question: 8 original trajectories and 8 length-augmented variants that retain the same underlying semantics. This design enables fine-grained analysis of model behavior under superficial variations in step length.

**Hyperparameter and Training Settings**  For joint training, we adopt a batch size of 128, with initial learning rates of $1 \times 10^{-4}$ for the PRM and $3 \times 10^{-4}$ for the bias estimator. The loss weights are set as $\lambda_r = 0.1$ and $\lambda_b = 0.5$ throughout training.

For experiments that train only the bias estimator, we use a batch size of 64. When training on the Math-Shepherd PRM, the initial learning rate is set to $2 \times 10^{-3}$, with $\lambda_{corr} = 0.3$. When using our own PRM (trained on PRM800K), we set the learning rate to $2 \times 10^{-3}$, also with $\lambda_{corr} = 0.3$.

We selected Qwen-2.5-Math-7B-instruct (Yang et al., 2024) as the foundational large language model (LLM) for our PRM, and Qwen-2.5-0.5B-instruct as the base model for the bias estimator. For joint training, four NVIDIA A100 GPUs were utilized, with the training process taking approximately 5 hours to complete. While solely training the bias estimator, only one RTX 4090 GPU was required, and this training procedure lasted around 3 hours. To enhance training resource efficiency, we employed Parameter-Efficient Fine-tuning techniques LoRA. The LoRA configuration was set with a rank of 8, an alpha value of 32, and dropout set to 0.1.

**Debiasing Methods in RLHF**

- **Length penalty(Singhal et al., 2023):** The method adds a linear proportional penalty term to the original reward
- **Loose Lips Sink Ships(Shen et al., 2023):** The method applies a Product-of-Experts framework that disentangles reward modeling from sequence-length effects by combining a semantic expert focused on human intent with a perturbed bias expert specialized in capturing length bias.
- **Adaptive Length Bias Mitigation(Bu et al., 2025):** The method introduces Adaptive Length Bias Mitigation (ALBM), which disentangles length bias from the original reward and adaptively recombines the length and quality rewards based on the characteristics of each query.
- **Uniform Average(Huang et al., 2024):** The method uses the local average reward to provide an estimation for the bias term, which can be removed, thereby approximating the true reward.
- **Locally Weighted Regression(LWR)(Huang et al., 2024):** The method assigns weights to nearby data points, giving higher importance to those closer to the target. This ensures that

proximate points significantly influence the regression. LWR then applies weighted linear regression to model the local behavior of the target function, effectively approximating the weighted average within the local context.

**Basemodels**

- **Math-Shepherd-PRM(Wang et al., 2023):** generates process labels by estimating the empirical probability that a step leads to the correct answer and trains a Process Reward Model (PRM) on their published dataset.
- **Base-PRM:** trained by ourselves using the human-annotated PRM800K dataset, based on Qwen-2.5-Math-7B-instruct (Yang et al., 2024).
- **Qwen2.5-Math-PRM(Zhang et al., 2025b):** adopts a two-phase data construction (data expansion, data filtering) and specific training. In expansion, it uses MC estimation with hard labels (a response is negative only if no 8 completions get the correct answer). In filtering, it uses Qwen2.5-Instruct-72B as LLM-as-a-judge to verify reasoning step-by-step, and applies consensus filtering to remove instances with mismatched LLM-annotated and MC-estimated process labels for quality. For training, it uses cross-entropy loss on end-of-step tokens for binary classification.
- **EurusPRM-Stage1(Cui et al., 2025):** is trained via Implicit PRM(Yuan et al., 2024), a framework that secures free process rewards without incurring additional costs—requiring only the simple training of an ORM (Outcome Reward Model) on more affordable response-level labels. During inference, implicit process rewards are generated by performing a forward pass and computing the log-likelihood ratio at each step.
- **ReasonFlux-PRM (Zou et al., 2025):**a trajectory-aware preference model that integrates step-level and trajectory-level supervision to provide fine-grained reward signals aligned with structured reasoning traces.
- **VersaPRM (Zeng et al., 2025):**a multi-domain PRM trained on synthetic reasoning data produced through novel data generation and annotation pipeline.

## C.2 EXAMPLE OF SEMI-SYNTHETIC SOLUTION

We generate extended variants either by duplicating the original step or by prompting DeepSeek to produce more verbose yet semantically equivalent rewrites, as illustrated in Figure 7. The prompt used for this process is provided in Figure 8.

## D SUPPLEMENTARY ABLATION STUDY

In this section, we present additional ablation results across different policy models, as shown in Table 5 and Table 6.

## E HYPERPARAMETER STUDY

### E.1 EFFECT OF $c$

We investigate the effect of the hyperparameter $c$, which controls the strength of the length bias correction in CoLD PRM. We evaluate the performance of CoLD PRM across different values of $c \in \{0.7, 0.9, 1.1, 1.3, 1.5\}$.

Figure 9 illustrates how varying $c$ affects model accuracy and solution length. The optimal $c$ varies when different models serve as policy models. For instance, when LLaMA-3-70B-Instruct acts as the policy model, it achieves the highest accuracy at $c = 1.1$; in contrast, when MetaMath-Mistral-7B functions as the policy model, it performs best at $c = 0.7$. Despite these discrepancies, a moderate value like $c = 1.1$ generally delivers strong results across all these policy models. This indicates that CoLD PRM is robust to the selection of $c$, and using a single $c$ value can effectively balance accuracy and solution length without the need for extensive per-model tuning.

**Question:** Convert the point $(0,3)$ in rectangular coordinates to polar coordinates. Enter your answer in the form $(r,\theta),$ where $r > 0$ and $0 \le \theta < 2 \pi.$

**Origin Version Solution:**

Step 1: Figure out $r.$ We have that $r = \sqrt{0^2 + 3^2} = 3.$

Step 2: Determine $\theta.$ The point $(0,3)$ lies on the positive $y$-axis, so this corresponds to an angle of $\frac{\pi}{2}$ radians (or 90 degrees). Thus, the polar coordinates are $\boxed{\left( 3, \frac{\pi}{2} \right).$

**Duplicating Version Solution:**

Step 1: Figure out $r.$ We have that $r = \sqrt{0^2 + 3^2} = 3.$ Figure out $r.$ We have that $r = \sqrt{0^2 + 3^2} = 3.$

Step 2: Determine $\theta.$ The point $(0,3)$ lies on the positive $y$-axis, so this corresponds to an angle of $\frac{\pi}{2}$ radians (or 90 degrees). Thus, the polar coordinates are $\boxed{\left( 3, \frac{\pi}{2} \right). Determine $\theta.$ The point $(0,3)$ lies on the positive $y$-axis, so this corresponds to an angle of $\frac{\pi}{2}$ radians (or 90 degrees). Thus, the polar coordinates are $\boxed{\left( 3, \frac{\pi}{2} \right).

**RewriteVersion Solution:**

Step 1: Determine the value of \( r \). Given the equation \( r = \sqrt{0^2 + 3^2} \), we simplify it to find that \( r = 3 \)

Step 2: To determine the angle $\theta$ in polar coordinates, we observe that the point $(0,3)$ is located on the positive $y$-axis. In the polar coordinate system, this position corresponds to an angle of $\frac{\pi}{2}$ radians (which is equivalent to 90 degrees). Therefore, the polar coordinates for this point are $\boxed{\left( 3, \frac{\pi}{2} \right)}$.

Figure 7: An example of the original and extended solutions

### E.2 EFFECT OF $\lambda_b$

We investigate the effect of the hyperparameter $\lambda_b$, which controls the strength of the correlation-based regularization in CoLD PRM. We evaluate the performance of CoLD PRM across different values of $\lambda_b \in \{0.3, 0.5, 0.7\}$.

Figure 10 illustrates how varying the value of $\lambda_b$ affects the model's accuracy and the length of selected solutions. It is evident that different values of $\lambda_b$ exhibit slight variations in performance across diverse datasets. Specifically, in certain cases, even when the length of the selected solution is relatively short, it may still result in a decline in accuracy. Consequently, an unwavering pursuit of brevity does not necessarily lead to the improvement of performance. Nevertheless, on the whole, CoLD PRM still demonstrates a certain degree of robustness.

### E.3 EFFECT OF $\alpha$

To further analyze potential over-penalization, we conduct additional experiments on the parameter $\alpha \in \{0.1, 0.01, 0.001, 0.0001, 0.00001\}$ across three Policy Model. From the Table 7, we can see that within a reasonable range of $\alpha$, CoLD demonstrates robustness, and the strength of the penalty controlled by $\alpha$ has very limited impact on performance. However, it is also clear that when $\alpha$

> **Prompt Used to Generate More Verbose but Semantically Equivalent Rewrites**
>
> I will provide you with a multi-step math problem solution. Your task is to rewrite each step by slightly expanding it while preserving its original meaning, without over-expanding. Please keep the 'Step i:' format for each step. The number of steps in your rewritten answer must match the original—do not split or add steps. If there are any mistakes in the original solution, retain them as they are and do not correct them.
>
> Below is the given solution process:
>     step1: ...
>     step2: ...
>     ...

Figure 8: Prompt Used to Generate More Verbose but Semantically Equivalent Rewrites by DeepSeek

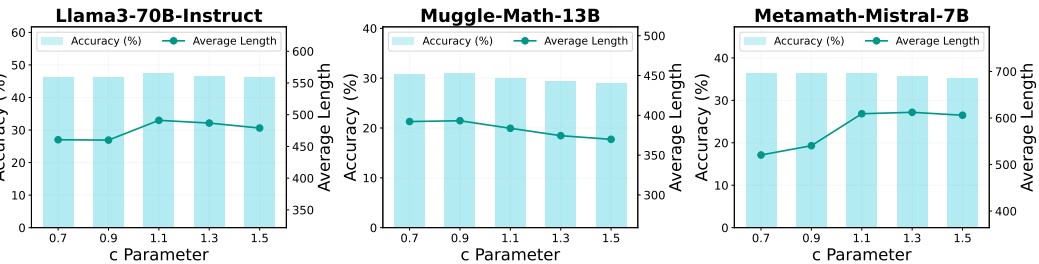

Figure 9: The performance of CoLD PRM under varying values of $c$ across different policy models.

applies an excessively strong penalty, over-penalization occurs. This indicates that over-penalization can appear in certain extreme cases.

## F  OTHER BENCHMARK

To better evaluate the capabilities of our model, we assess it on ProcessBench (Zheng et al., 2024)—a public benchmark for Process Reward Models (PRMs). The goal of this evaluation is to determine whether the PRM can identify the first erroneous step in the reasoning process. ProcessBench partitions the dataset into two subsets: one containing samples with incorrect final answers and the other with correct final answers. It then computes the harmonic mean of the accuracies achieved on these two subsets to obtain the final F1-score.

As shown in the Table 8, our CoLD PRM outperforms nearly all open-source PRM baselines across all datasets. The performance advantage is particularly notable on the more challenging datasets, including OlympiadBench and OmniMATH, demonstrating that our CoLD PRM maintains strong generalization over OOD datasets.

When comparing models across different scales, CoLD still maintains strong competitiveness. In terms of overall performance, it is even able to outperform some larger-scale counterpart models.

## G  EVALUATION USING STRONGER POLICY MODEL

In addition to the policy models reported in the main paper, we further include Qwen3-Next-80B-A3B-Instruct Yang et al. (2025) and DeepSeek-v3.1 Liu et al. (2024a) in our evaluation. These

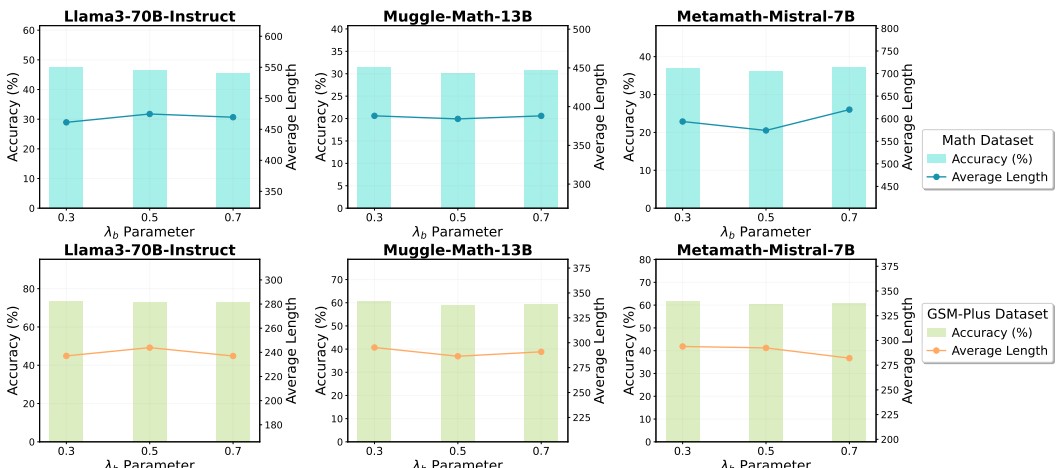

Figure 10: The performance of CoLD PRM under varying values of $\lambda_b$ across different policy models.

Table 7: Effect of different $\alpha$ values on CoLD performance across policy models.

| Policy Model | $\alpha$ | MATH500 | | GSM-Plus | | Avg | |
|---|---|---|---|---|---|---|---|
| | | ArithACC(%) | Length | ArithACC(%) | Length | ArithACC(%) | Length |
| Llama-3-70B-Instruct | 0.1 | 38.8 | 222.6 | 68.6 | 177.4 | 53.7 | 200.0 |
| | 0.01 | 42.4 | 223.5 | 71.1 | 179.6 | 56.8 | 201.6 |
| | 0.001 | 48.6 | 253.5 | 72.6 | 189.7 | 60.6 | 221.6 |
| | 0.0001 | 49.2 | 313.2 | 73.8 | 202.5 | 61.5 | 257.9 |
| | 0.00001 | 47.9 | 350.6 | 73.3 | 210.4 | 60.6 | 280.5 |
| MetaMath-Mistral-7B | 0.1 | 25.0 | 215.3 | 45.5 | 202.9 | 35.3 | 209.1 |
| | 0.01 | 31.2 | 218.9 | 50.2 | 205.3 | 40.7 | 212.1 |
| | 0.001 | 38.8 | 276.29 | 56.9 | 220.2 | 47.9 | 248.2 |
| | 0.0001 | 37.2 | 376.3 | 61.4 | 238.6 | 49.3 | 307.5 |
| | 0.00001 | 36.2 | 393.8 | 59.9 | 259.4 | 48.1 | 326.6 |
| Muggle-Math-13B | 0.1 | 21.4 | 191.1 | 43.7 | 193.1 | 32.6 | 192.1 |
| | 0.01 | 27.0 | 195.9 | 50.0 | 195.8 | 38.5 | 195.9 |
| | 0.001 | 30.2 | 248.4 | 58.5 | 215.8 | 44.4 | 232.1 |
| | 0.0001 | 31.4 | 309.2 | 60.3 | 243.3 | 45.9 | 276.3 |
| | 0.00001 | 30.2 | 357.7 | 59.62 | 263.2 | 44.9 | 310.4 |

models represent state-of-the-art large-scale language models that have demonstrated substantially stronger reasoning capabilities compared with the earlier baselines.

As shown in the Table **??** , our CoLD framework continues to deliver clear and consistent improvements even when applied to these more advanced LLMs. This confirms that CoLD is not merely compensating for weaknesses in smaller models, but instead provides generalizable gains that hold across diverse policy models, including the latest high-performing LLMs.

# H  BIAS ESTIMATOR INTERPRETABILITY

To better understand what the Bias Estimator learns and how it makes predictions, we conduct interpretability analyses focusing on its sensitivity to input length versus semantic content. Through a combination of output case studies and attention-based comparisons, we aim to clarify whether the Bias Estimator captures spurious surface-level cues, specifically length, rather than the underlying meaning of the input.

Table 8: The performance of different models on ProcessBench. The best result is given in **bold**, and the second-best value is underlined.

| | Model | GSM8k | | MATH | | OlympiadBench | | OmniMATH | | Avg.F1 |
|---|---|---|---|---|---|---|---|---|---|---|
| | | ArithACC | F1 | ArithACC | F1 | ArithACC | F1 | ArithACC | F1 | |
| Open-source PRM | CoLD-Base-PRM-7B(Ours) | 72.0 | 68.6 | **67.3** | **67.7** | 54.6 | 56.0 | 47.8 | 51.3 | **60.9** |
| | Qwen2.5-Math-7B-PRM800K | **73.5** | 68.2 | 65.1 | 62.6 | 53.2 | 50.7 | 43.4 | 44.3 | 56.5 |
| | Skywork-PRM-7B | 71.6 | **70.8** | 54.5 | 53.6 | 25.6 | 22.9 | 23.7 | 21.0 | 42.1 |
| | Skywork-PRM-1.5B | 59.9 | 59.0 | 49.1 | 48.0 | 20.5 | 19.3 | 19.7 | 19.2 | 36.4 |
| | Math-Shepherd-PRM-7B | 58.3 | 47.9 | 45.1 | 29.5 | 39.7 | 24.8 | 34.8 | 23.8 | 31.5 |
| | RLHFlow-PRM-Mistral-8B | 62.3 | 50.4 | 42.1 | 33.4 | 22.3 | 13.8 | 19.1 | 15.8 | 28.4 |
| | RLHFlow-PRM-Deepseek-8B | 56.9 | 38.8 | 45.1 | 33.8 | 26.5 | 16.9 | 23.2 | 16.9 | 26.6 |
| Language Models as Critic | QwQ-32B-Preview | **87.9** | **88.0** | **78.5** | **78.7** | 59.2 | **57.8** | **61.1** | **61.3** | **71.5** |
| | GPT-4o | 80.2 | 79.2 | 63.4 | 63.6 | 50.1 | 51.4 | 50.1 | 53.5 | 61.9 |
| | Qwen2.5-72B-Instruct | 77.9 | 76.2 | 65.4 | 61.8 | **59.8** | 54.6 | 55.1 | 52.2 | 61.2 |
| | Llama-3.3-70B-Instruct | 83.7 | 82.9 | 63.7 | 59.4 | 54.3 | 46.7 | 51.0 | 43.0 | 58.0 |
| | Qwen2.5-Coder-32B-Instruct | 72.0 | 68.9 | 64.5 | 60.1 | 57.0 | 48.9 | 52.5 | 46.3 | 56.1 |
| | Llama-3.1-70B-Instruct | 75.3 | 74.9 | 52.6 | 48.2 | 50.0 | 46.7 | 43.2 | 41.0 | 52.7 |
| | Qwen2.5-14B-Instruct | 72.3 | 69.3 | 59.2 | 53.3 | 50.2 | 45.0 | 43.5 | 41.3 | 52.2 |
| | Qwen2-72B-Instruct | 67.8 | 67.6 | 52.3 | 49.2 | 43.3 | 42.1 | 39.3 | 40.2 | 49.8 |
| | Qwen2.5-32B-Instruct | 70.6 | 65.6 | 61.9 | 53.1 | 53.5 | 40.0 | 47.7 | 38.3 | 49.3 |
| | Qwen2.5-Math-72B-Instruct | 70.3 | 65.8 | 59.6 | 52.1 | 56.1 | 32.5 | 55.1 | 31.7 | 45.5 |
| | Qwen2.5-Coder-14B-Instruct | 61.9 | 50.1 | 54.2 | 39.9 | 51.4 | 34.0 | 55.6 | 27.3 | 37.8 |
| | Qwen2.5-7B-Instruct | 37.8 | 36.5 | 36.9 | 36.6 | 29.9 | 29.7 | 27.3 | 27.4 | 32.6 |
| | Meta-Llama-3-70B-Instruct | 62.4 | 52.2 | 48.3 | 22.8 | 46.2 | 21.2 | 44.8 | 20.0 | 29.1 |
| | Qwen2.5-Math-7B-Instruct | 54.4 | 26.8 | 50.3 | 25.7 | 43.1 | 14.2 | 41.6 | 12.7 | 19.9 |
| | Qwen2-7B-Instruct | 25.1 | 8.4 | 20.4 | 19.0 | 16.1 | 14.7 | 13.8 | 12.1 | 13.6 |
| | Meta-Llama-3-8B-Instruct | 27.1 | 13.1 | 17.3 | 13.8 | 14.2 | 4.8 | 19.7 | 12.6 | 11.1 |
| | Qwen2.5-Coder-7B-Instruct | 49.1 | 14.3 | 46.3 | 6.5 | 47.2 | 4.1 | 48.9 | 1.8 | 6.7 |
| | Llama-3.1-8B-Instruct | 27.3 | 10.9 | 20.5 | 5.1 | 16.0 | 2.8 | 15.0 | 1.6 | 5.1 |

Table 9: Evaluation Results of CoLD Using Advanced Large-Scale Policy Models on MATH500.

| Policy Model | PRM model | ArithACC(%) | Length |
|---|---|---|---|
| Qwen3-Next-80B -A3B-Instruct | Vanilla Base PRM | 85.2 | 1050.6 |
| | CoLD Base PRM(Ours) | 90.4 | 754.5 |
| Deepseek-v3.1 | Vanilla Base PRM | 89.6 | 953.9 |
| | CoLD Base PRM(Ours) | 91.6 | 646.2 |

## H.1 CASE STUDY

To illustrate the interpretability of the Bias Estimator, we present a case study demonstrating its output behavior. As shown in Figure 11, the Bias Estimator responds more strongly to length than to semantic content. For example, in Step 3 of the two answers displayed, although their semantic meaning differs and one step is correct while the other is incorrect, the Bias Estimator assigns similar scores because it is primarily driven by step length rather than semantics.

## H.2 ATTENTION VISUALIZATION

- Sentence A: Compute $r$. Here $x = 0$ and $y = 3$, so $r = \sqrt{0^2 + 3^2} = \sqrt{9} = 3$.

- Sentence B: Analyze $k$. If a machine runs 12 cycles in 4 seconds, then $k = 12/4 = 3$.

To examine whether the Bias Estimator relies on length rather than semantic information, we conduct two comparative attention analyses. First, we feed Sentence A into both the Bias Estimator and the backbone model and compare their attention distributions. As shown in Figure 12, while the backbone exhibits semantically focused attention peaks, the Bias Estimator assigns attention almost uniformly across tokens, without concentrating on semantic-bearing words. Second, we input Sentence A and Sentence B, which share the same length but differ completely in semantics, into the Bias Estimator. As illustrated in Figure 13, their attention maps remain nearly indistinguishable, indicating that the attention pattern remains stable when semantics change while length is kept constant. Together, these results reinforce that the Bias Estimator's attention primarily encodes length-related cues rather than semantic information.

---

**Question:** Convert the point $(0,3)$ in rectangular coordinates to polar coordinates. Enter your answer in the form $(r,\theta),$ where $r > 0$ and $0 \le \theta < 2 \pi.$

---

**Answer1:**

**Step 1.** Use the formulas $(r,\theta)$ satisfy $r = \sqrt{x^2 + y^2}, \theta = \operatorname{atan2}(y,x)$ (0.2806)

**Step 2.** Compute $r$. Here $x=0$ and $y=3$, so $r = \sqrt{0^2 + 3^2} = \sqrt{9} = 3.$ (0.3257)

**Step 3.** Determine $\theta$. The point $(0,3)$ is on the positive $y$-axis, so $\theta = \frac{\pi}{2}$. (0.3541)

**Step 4.** State the final answer:$(0,3) \longrightarrow (3, \tfrac{\pi}{2}).$ (0.2556)

---

**Answer2:**

**Step 1.** Use the formulas $(r,\theta)$ satisfy $r = \sqrt{x^2 + y^2}, \theta = \operatorname{atan2}(y,x)$ (0.2806)

**Step 2.** Compute $r$. Here $x=0$ and $y=3$, so $r = \sqrt{0^2 + 3^2} = \sqrt{9} = 3.$ (0.3257)

**Step 3.** Determine $\theta$. Using $\theta = \arctan\left(\frac{y}{x}\right)$ gives $\theta = \arctan\left(\frac{3}{0}\right) = 0$. (0.3491)

**Step 4.** State the final answer:$(0,3) \longrightarrow (3, 0)$.(0.2456)

---

Figure 11: Case study of the Bias Estimator outputs, where the red numbers in parentheses at the end of each step denote the predicted scores.

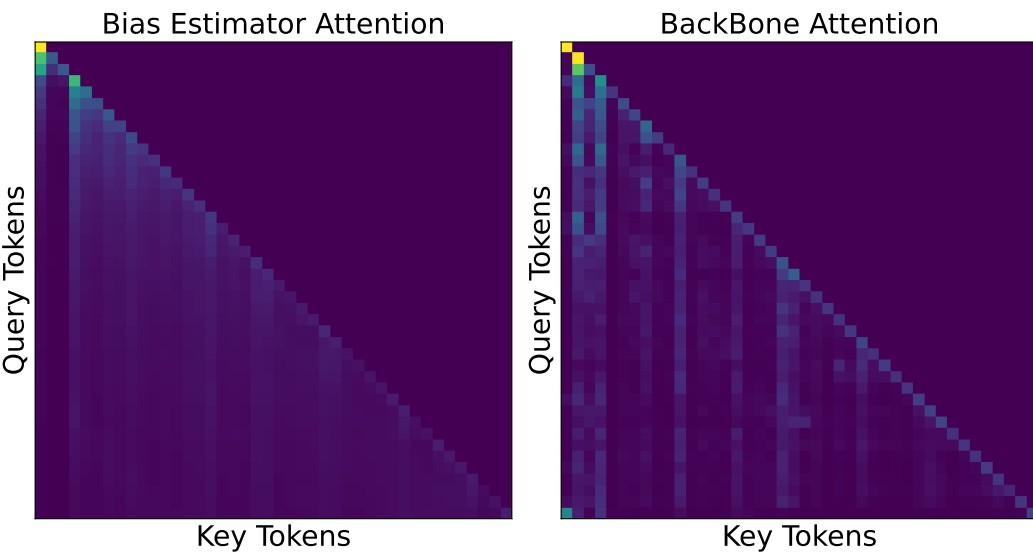

Figure 12: Attention visualizations for Sentence A in the backbone model and the Bias Estimator.

## LARGE LANGUAGE MODELS (LLMS) USAGE

In this work, large language models (LLMs) were used solely for text polishing and language refinement. They were not involved in the design of the methodology, implementation, analysis, or the generation of experimental results. All technical contributions and research findings are entirely the work of the authors.

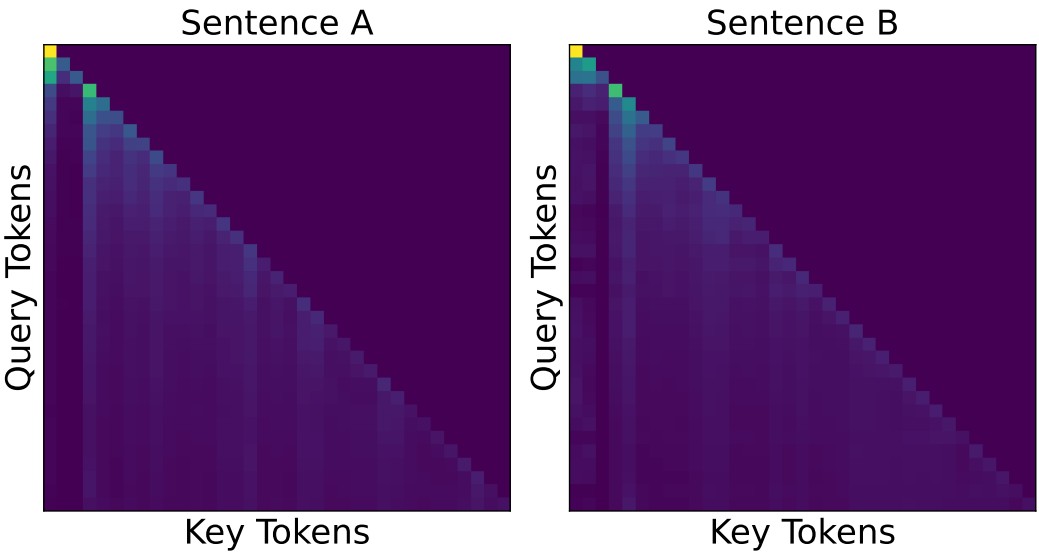

Figure 13: Attention visualizations for Sentence A and Sentence B, which share identical length but differ in semantics, under the Bias Estimator.

