# OpenReview forum: "CoLD: Counterfactually-Guided Length Debiasing for Process Reward Models in Mathematical Reasoning"
_ICLR.cc/2026/Conference — Submitted to ICLR 2026_

### Official Review · Reviewer_pjZB · 2025-10-15

**Soundness:** 4
**Presentation:** 2
**Contribution:** 2
**Rating:** 4
**Confidence:** 4

**Summary:**

This paper addresses the critical issue of 'length bias' in PRMs where verbose reasoning steps are incorrectly favored. The authors introduce CoLD, a novel framework that uses a causal inference perspective to model and mitigate this bias. CoLD employs a dedicated bias estimator and a specialized joint training objective to disentangle the true reward for logical correctness from spurious, length-related artifacts. Extensive experiments on several math reasoning benchmarks demonstrate that the proposed CoLD framework significantly improves Best-of-N style TTS evaluation, offering a robust solution to a key challenge in process reward modeling.

**Strengths:**

*   **Novel principled causal formulation:** The paper's innovation lies on its reframing of length bias through a causal lens. It formally models step length as a confounder and establishes a counterfactual objective to isolate semantic correctness from spurious correlations. This provides a strong theoretical foundation that elevates the work beyond prior heuristic-based debiasing approaches.

*   **Modular architecture and extensive ablation:** CoLD features a highly modular architecture that enhances its practical utility. The extensive ablation studies compellingly demonstrate that its components.

* **Consistent Improvement**: Experiments show that CoLD consistently achieves test-time scaling (TTS) improvements.

**Weaknesses:**

* **Extra computational cost**: CoLD introduces substantial additional computational overhead, yet it does not yield compelling performance gains—especially given that the paper evaluates using Best-of-N, a high-inference-cost strategy. This severely limits its practical applicability.

* **Limited generalization**: Results in Tables 1 and 2 show that the method performs more noticeably on larger LLMs, which may be attributed to their inherent reasoning diversity. The marginal improvements on smaller models reveal weak generalization capability.

* **Online test**: Works such as EurusPRM and DeepSeekMath have already explored integrating PRMs with online methods like GRPO and reinforcement learning, and the current paper offers insights into length bias. Why does it not investigate how length bias affects online PRM utilization or the benefits of CoLD in such dynamic settings? Addressing this would significantly enhance CoLD’s practical value, whereas the current approach remains confined to static test-time scaling.

* **Baselines**: More advanced LLMs should be considered: Models such as Qwen3 should be included in the evaluation. For instance, the chosen baselines already exhibit relatively low performance on Math500 under current research standards.

**Questions:**

Please see weaknesses.

---

> ### Author Response · Authors · 2025-11-21
> **Repsonse to Reviewer pjZB (Part 1)**
>
> >Extra computational cost: CoLD introduces substantial additional computational overhead, yet it does not yield compelling performance gains—especially given that the paper evaluates using Best-of-N, a high-inference-cost strategy. This severely limits its practical applicability.
>
> We appreciate the reviewer’s concern regarding computational cost. **CoLD adds only a lightweight auxiliary model, and the resulting overhead is therefore quite limited.** To verify this, we conducted a runtime analysis on 500 questions and compared the total inference time between the vanilla system and the CoLD-augmented system. The results are presented in the table below and show that **the additional latency introduced by CoLD is minimal relative to the overall pipeline.**
>
> | Method                       | Time Consumption(s) |
> | ---------------------------- | ------------------- |
> | PRM-only(7B)                 | 184                 |
> | PRM(7B)+Bias Estimator(0.5B) | 200                 |
>
> Regarding the comment on the high cost of Best-of-N, we would like to clarify that **Best-of-N is a test-time scaling strategy used during evaluation rather than a component of our method. The computational burden associated with Best-of-N is independent of CoLD itself**. Moreover, prior work [1][2][3][4] also uses Best-of-N for evaluation, which makes its use consistent with standard practice in PRM-based reasoning.
>
> In addition to Best-of-N evaluation, we further assess CoLD on ProcessBench[5], a widely used benchmark for evaluating PRMs. As reported in Appendix F, the ProcessBench results also confirm the effectiveness of CoLD, demonstrating that its benefits are not limited to Best-of-N settings.
>
>
> [1] Li W, Li Y. Process Reward Model with Q-Value Rankings
>
> [2] Zou J, Yang L, Gu J, et al. ReasonFlux-PRM: Trajectory-Aware PRMs for Long Chain-of-Thought Reasoning in LLMs
>
> [3] Sun L, Liu C, Ma X, et al.  FreePRM: Training Process Reward Models Without Ground Truth Process Labels
>
> [4] Liu Y, Lu J, Chen Z, et al. Adaptivestep: Automatically dividing reasoning step through model confidence
>
> [5] Zheng C, Zhang Z, Zhang B, et al. Processbench: Identifying process errors in mathematical reasoning
>
> > Limited generalization: Results in Tables 1 and 2 show that the method performs more noticeably on larger LLMs, which may be attributed to their inherent reasoning diversity. The marginal improvements on smaller models reveal weak generalization capability.
>
> We appreciate the reviewer’s observation regarding the relative improvements across different model scales. **It is indeed expected that larger LLMs exhibit greater reasoning diversity, which allows them to benefit more substantially from debiasing methods such as CoLD.**
>
> At the same time, we would like to emphasize that **CoLD still brings consistent improvements to smaller models across all evaluated datasets.** Although the absolute gains are modest, the direction of improvement is uniform, indicating that **the method generalizes across model sizes rather than being restricted to high-capacity LLMs.** Moreover, smaller models inherently have limited semantic expressiveness, which restricts the magnitude of improvement that any debiasing method can introduce, especially when the PRM itself has limited resolution in distinguishing subtle reasoning differences.
>
> Moreover, as model capabilities continue to improve and larger models become the norm, **CoLD is therefore positioned to deliver even more substantial benefits on the stronger models** that are driving current and future progress in reasoning tasks.

---

> ### Author Response · Authors · 2025-11-21
> **Repsonse to Reviewer pjZB (Part 2)**
>
> >Online test: Works such as EurusPRM and DeepSeekMath have already explored integrating PRMs with online methods like GRPO and reinforcement learning, and the current paper offers insights into length bias. Why does it not investigate how length bias affects online PRM utilization or the benefits of CoLD in such dynamic settings? Addressing this would significantly enhance CoLD’s practical value, whereas the current approach remains confined to static test-time scaling.
>
> Thank you for raising this important point. We agree that understanding how length bias affects online PRM utilization in reinforcement learning is crucial for assessing the practical value of debiasing methods.
>
> To more comprehensively demonstrate CoLD’s effectiveness on downstream tasks, **we apply CoLD in a reinforcement learning setting.** Specifically, our experiments are conducted within the ReasonFlux-PRM framework[1]. We use Qwen2.5-Math-1.5B-Instruct as the policy model and adopt GRPO (Group Relative Policy Optimization)[2] as the RL optimization algorithm. CoLD is integrated into the ReasonFlux-PRM and compared against its vanilla counterpart. For evaluation, we use MATH500 as the test benchmark. The results are summarized below.
>
>
> | Model                                             | Accuracy | Length |
> | ------------------------------------------------- | -------- | ------ |
> | Qwen2.5-Math-1.5B-Instruct+Vanilla ReasonFlux-PRM | 54.4     | 600.9  |
> | Qwen2.5-Math-1.5B-Instruct+CoLD ReasonFlux-PRM    | 57.6     | 543.2  |
>
> As shown in the table above, incorporating CoLD not only improves the accuracy of the policy model in answering questions, but also reduces the output length. This indicates that **CoLD effectively mitigates the length bias exhibited by the PRM, providing the model with more precise and meaningful reward signals during downstream reinforcement learning training.** Our RL experiments confirm that CoLD enhances PRM behavior not only at test time scaling but also during dynamic learning, where biased rewards would otherwise propagate and amplify over training steps.
>
>
> [1] Zou J, Yang L, Gu J, et al. ReasonFlux-PRM: Trajectory-Aware PRMs forLong Chain-of-Thought Reasoning in LLMs
>
> [2] Shao Z, Wang P, Zhu Q, et al. DeepSeekMath: Pushing the Limits of Mathematical Reasoning in Open Language Models
>
> >Baselines: More advanced LLMs should be considered: Models such as Qwen3 should be included in the evaluation. For instance, the chosen baselines already exhibit relatively low performance on Math500 under current research standards.
>
> Thank you for the insightful comment. We agree that evaluating on stronger LLMs is important for demonstrating the robustness of our approach. In addition to the models reported in the main paper, we have further included Qwen3-Next-80B-A3B-Instruct[1] and DeepSeek-v3.1[2], both of which represent state-of-the-art large-scale models with substantially stronger reasoning capabilities.
>
> As shown in the table below, **our CoLD framework continues to deliver clear and consistent improvements even when applied to these more advanced LLMs.** This confirms that CoLD is not merely compensating for weaknesses in smaller models, but instead provides generalizable gains that hold across diverse policy models, including the latest high-performing LLMs.
>
> **Policy Model:Qwen3-Next-80B-A3B-Instruct**
> | Alpha          | ArithACC(MATH500) | Length(MATH500) |
> | -------------- | ----------------- | --------------- |
> | Vanilla PRM    | 85.2              | 1050.6          |
> | CoLD(Ours) PRM | 90.4              | 754.5           |
>
>
> **Policy Model:Deepseek-v3.1**
> | Alpha          | ArithACC(MATH500) | Length(MATH500) |
> | -------------- | ----------------- | --------------- |
> | Vanilla PRM    | 89.6              | 953.9           |
> | CoLD(Ours) PRM | 91.6              | 646.2           |
>
>
> [1] Yang A, Li A, Yang B, et al. Qwen3 technical report
>
> [2] Liu A, Feng B, Xue B, et al. Deepseek-v3 technical report

---

> ### Author Response · Authors · 2025-11-27
>
> Dear Reviewer pjZB,
>
> I hope this message finds you well. We have carefully addressed the questions and concerns you previously raised, and we would greatly appreciate any further feedback you might be able to provide. Your insights are highly valuable to us, and we remain fully available to clarify or refine any remaining points.
>
> We sincerely appreciate your response at your earliest convenience, which will greatly assist us to move the discussion forward.
>
> Thank you sincerely for your time and effort in reviewing our paper.

---

### Official Review · Reviewer_eaM8 · 2025-10-24

**Soundness:** 2
**Presentation:** 2
**Contribution:** 1
**Rating:** 2
**Confidence:** 4

**Summary:**

This paper examines the issue of length bias in PRMs for mathematical reasoning, where longer reasoning steps receive higher reward scores regardless of logical validity. The authors propose CoLD, a framework composed of three parts: a length-penalty term, a learnable bias estimator network, and a joint training procedure with correlation regularization to separate semantic correctness from superficial verbosity. Experiments on MATH500 and GSM-Plus show slightly higher accuracy and shorter generated reasoning steps compared with baseline PRMs.

**Strengths:**

- The experimental setup is reproducible, and datasets are public.

**Weaknesses:**

1. The method does not implement true counterfactual inference or causal adjustment; it is a multi-loss regularization scheme.

2. The paper’s central idea "jointly learning a bias estimator with a reward model to remove spurious correlation with response length" appears conceptually similar to several prior studies addressing length bias in reward modeling.

- **Shen et al. (2023)**, *Loose Lips Sink Ships: Mitigating Length Bias in RLHF* (arXiv: [2310.05199](https://arxiv.org/abs/2310.05199)), separates a “length expert” and a “reward expert” under a Product-of-Experts formulation to decouple length bias. CoLD’s “bias estimator + main PRM joint training” follows a related structure, though applied to mathematical reasoning.

- **Huang et al. (2024)**, *Post-hoc Reward Calibration: A Case Study on Length Bias* (arXiv: [2409.17407](https://arxiv.org/abs/2409.17407)), estimates a length-bias function and subtracts it from the reward, which is mathematically close to CoLD’s formulation
\[
r^*(x) = r_\theta(x) - c·σ(b_φ(x)) - αℓ(x)
\]
but achieved without retraining.

- **Bu et al. (2025)**, *Adaptive Length Bias Mitigation in Reward Models for RLHF* (Findings of ACL NAACL 2025, [PDF](https://aclanthology.org/2025.findings-naacl.169.pdf)), introduces a dynamic penalty for verbosity bias that pursues a similar goal from a simpler perspective.

  While CoLD extends these ideas to process reward models, the paper does not clearly articulate its methodological distinctions or provide empirical comparisons with these baselines. This potential overlap weakens the perceived novelty and limits the strength of the claimed contribution.

3. No justification of parameterization or stability (Line 825). The hyper-parameters play critical roles but are not theoretically motivated.The paper provides no sensitivity study or guidance for setting them, making reproducibility fragile. A method presented as “causally principled” should not rely on manual tuning to achieve the effect.

4. Limited scope of evaluation. The experimental evaluation is narrow and lacks strong baselines.
All experiments are conducted only on math-specific reasoning datasets (MATH500 and GSM-Plus), without testing on other reasoning or RLHF-related settings. Moreover, the paper compares CoLD only with simple internal variants and does not include key external baselines. Without such comparisons, it is difficult to assess whether CoLD offers any meaningful improvement over existing methods or if its gains merely result from additional regularization.

**Questions:**

Refer to Weaknesses.

---

> ### Author Response · Authors · 2025-11-21
> **Repsonse to Reviewer eaM8 (Part 1)**
>
> > The method does not implement true counterfactual inference or causal adjustment; it is a multi-loss regularization scheme.
>
> We appreciate the reviewer’s thoughtful observation. We respectfully clarify that **our method is indeed grounded in counterfactual reasoning, and the use of multi-loss regularization does not contradict this.** Specifically, Eq. 3 corresponds exactly to the counterfactual effect:
>
> $$
> E_{C,N \rightarrow P} = P(C=c,N=n,L=\ell) - P(C=c^\star,N=n^\star,L=\ell),
> $$
>
> which isolates the causal contribution of correctness while holding step length fixed. Each term has a clear semantic interpretation: the composed reward $r_\theta(x)\sigma(b_\phi(x+\mathcal{N}))$ estimates the factual quantity $P(C=c,N=n,L=\ell)$, while $\sigma(b_\phi(x+\mathcal{N}))$ and $\alpha \ell(x)$ approximate the counterfactual baseline $P(C=c^\star,N=n^\star,L=\ell)$. This structure arises directly from the algebra of counterfactual inference, not from heuristic trial-and-error.
>
> **The multi-loss objective is a practical means to implement this theory.** Its role is to ensure that the bias estimator effectively captures the step length feature, thereby allowing the PRM to separate the influence of length on the predicted score. In other words, the loss is designed to facilitate accurate estimation of the components required by the counterfactual decomposition. The causal theory informs the derivation of the inference formula (Eq. 3), while the loss provides a concrete mechanism to realize it in practice. Thus, **our approach does implement counterfactual reasoning, and the multi-loss regularization is a complementary tool.**

---

> ### Author Response · Authors · 2025-11-21
> **Repsonse to Reviewer eaM8 (Part 2)**
>
> >The paper’s central idea "jointly learning a bias estimator with a reward model to remove spurious correlation with response length" appears conceptually similar to several prior studies addressing length bias in reward modeling.
>
> > While CoLD extends these ideas to process reward models, the paper does not clearly articulate its methodological distinctions or provide empirical comparisons with these baselines. This potential overlap weakens the perceived novelty and limits the strength of the claimed contribution.
>
> We thank the reviewer for pointing this out and for the thoughtful comment regarding the novelty of our work relative to existing methods. While prior work has largely focused on outcome-level scoring, our work specifically addresses PRMs, which operate on step-level reasoning and are sensitive to the sequential structure of intermediate steps.
>
> To better illustrate that traditional reward model designs may not be suitable for the PRM setting, **we compared several debiasing methods originally developed for traditional reward model.** In the table below, LLSS is the abbreviation for Loose Lips Sink Ships, which corresponds to the method in paper [1]; ALBM stands for Adaptive Length Bias Mitigation, the method proposed in paper [2]; UA denotes Uniform Average, and LWR stands for Locally Weighted Regression, the method from paper [3]. I have already compared [3] as a baseline in Table 1 of the paper, and it is provided again here for convenient direct comparison.
>
> **Policy Model:llama3-70b-instruct**
> | Method     | ArithACC(MATH500) | Length(MATH500) | ArithACC(GSM-Plus) | Length(GSM-Plus) |
> | ---------- | ----------------- | --------------- | ------------------ | ---------------- |
> | LLSS       | 45.8              | 435.7           | 72.5               | 302.5            |
> | ALBM       | 43.0              | 594.8           | 72.2               | 364.3            |
> | UA         | 40.4              | 608.5           | 70.4               | 324.6            |
> | LWR        | 39.0              | 590.7           | 69.6               | 299.9            |
> | CoLD(Ours) | 49.2              | 313.2           | 73.8               | 202.5            |
>
>
>
>
> **Policy Model:Muggle-Math-13B**
> | Method      | ArithACC(MATH500) | Length(MATH500) | ArithACC(GSM-Plus) | Length(GSM-Plus) |
> | ---------- | ----------------- | --------------- | ------------------ | ---------------- |
> | LLSS       | 28.6              | 375.7           | 59.1               | 263.9            |
> | ALBM       | 25.4              | 472.0           | 56.3               | 364.5            |
> | UA         | 27.4              | 440.8           | 55.3               | 328.7            |
> | LWR        | 23.5              | 470.5           | 50.7               | 312.3            |
> | CoLD(Ours) | 31.4              | 309.2           | 60.3               | 243.3            |
>
>
> **Policy Model:MetaMath-Mistral-7B**
> | Method      | ArithACC(MATH500) | Length(MATH500) | ArithACC(GSM-Plus) | Length(GSM-Plus) |
> | ---------- | ----------------- | --------------- | ------------------ | ---------------- |
> | LLSS       | 34.5              | 441.2           | 58.0               | 340.0            |
> | ALBM       | 31.0              | 610.0           | 56.5               | 385.0            |
> | UA         | 33.2              | 609.9           | 56.3               | 363.8            |
> | LWR        | 29.6              | 507.3           | 52.7               | 354.4            |
> | CoLD(Ours) | 37.2              | 376.3           | 61.4               | 238.6            |
>
>
> From the experimental results above, we observe that although these methods effectively address length bias in traditional reward modeling, **they perform noticeably worse in the PRM setting**, exhibiting subpar results in both accuracy and the length of the selected responses. In contrast, our CoLD method successfully mitigates the length bias that emerges in PRM.
>
> Meanwhile, **Our method is derived from causal-theoretic principles (with detailed derivations provided in Appendix A)**, whereas their approach is largely intuitive and lacks rigorous theoretical justification. Moreover, **we are the first to apply this framework in the PRM setting**, which involves step-level scoring. Compared to the traditional reward model paradigm, this task is considerably more complex, requiring a more careful and nuanced design and implementation.
>
> [1] Shen W, Zheng R, Zhan W, et al. Loose lips sink ships: Mitigating length bias in reinforcement learning from human feedback
>
> [2] Bu Y, Huo L, Jing Y, et al. Adaptive Length Bias Mitigation in Reward Models for RLHF
>
> [3] Huang Z, Qiu Z, Wang Z, et al. Post-hoc Reward Calibration: A Case Study on Length Bias

---

> > ### Author Response · Authors · 2025-11-21
> > **Repsonse to Reviewer eaM8 (Part 3)**
> >
> > > No justification of parameterization or stability (Line 825). The hyper-parameters play critical roles but are not theoretically motivated.The paper provides no sensitivity study or guidance for setting them, making reproducibility fragile. **A method presented as “causally principled” should not rely on manual tuning to achieve the effect.**
> >
> >
> > We thank the reviewer for raising this concern. **As documented in Appendix E, we have already conducted a comprehensive hyperparameter study in which we systematically explore the ranges of key hyperparameters.** The results show that our method is highly **robust**: even when the chosen hyperparameter settings deviate from the optimal configuration, the model maintains strong performance across the main evaluation metrics. This indicates that our approach does not rely on fragile or heavily tuned hyperparameters, and its effectiveness is stable over a broad parameter range.
> >
> > Importantly, **we would like to emphasize that the causal nature of CoLD does not rely on hyperparameter tuning.** The hyperparameters only determine the relative strength with which the model enforces the causal decomposition during optimization. They do not alter the causal structure itself. This situation is entirely analogous to well-established causal methods such as inverse propensity weighting, causal regularization, and variational front-door adjustments, where weighting coefficients are routinely introduced to balance causal components during training. **The presence of such coefficients has never been viewed as conflicting with the causal validity of these methods, because the coefficients modulate optimization dynamics rather than define the causal effect.**
> >
> > Similarly, in CoLD, the identification of the causal components $E_{L \rightarrow P}$ and $E_{C,N \rightarrow P}$ is fully determined by the model architecture and the factorization assumptions, not by the specific value of any hyperparameter. Tuning affects only the extent to which the training process emphasizes debiasing relative to other learning signals. **Therefore, the causal justification behind CoLD does not become weaker due to the existence of hyperparameters, and the method does not depend on fine-grained manual tuning to realize its intended causal effect.**
> >
> > > Limited scope of evaluation. The experimental evaluation is narrow and lacks strong baselines. All experiments are conducted only on math-specific reasoning datasets (MATH500 and GSM-Plus), without testing on other reasoning or RLHF-related settings. Moreover, the paper compares CoLD only with simple internal variants and does not include key external baselines. Without such comparisons, it is difficult to assess whether CoLD offers any meaningful improvement over existing methods or if its gains merely result from additional regularization.
> >
> > We sincerely appreciate the reviewer’s thoughtful comments regarding the scope of our evaluation. We would like to clarify that our experimental setup is not limited to a narrow or single-setting benchmark. To address the concern directly, **we outline below both the experiments already included in the paper and the additional ones we have incorporated.**
> >
> > - **ProcessBench** is a benchmark designed to assess the ability to detect incorrect steps in mathematical reasoning. It contains 3,400 test cases, primarily drawn from competition- and Olympiad-level problems. As shown in Appendix F, **our CoLD-based PRM consistently outperforms nearly all open-source PRM baselines across the full suite of datasets.** The performance gap is especially pronounced on the more challenging benchmarks—such as OlympiadBench and OmniMATH, highlighting that CoLD-PRM maintains strong generalization even under significant distribution shifts.
> > - **Comparison with other baselines**. In Table 1 of the paper, we have already compared CoLD with two baselines[5]. To more comprehensively demonstrate CoLD’s effectiveness, we further include comparisons with two additional baselines[3][4]. The experimental results, presented in the discussion above, show that **CoLD continues to exhibit strong performance relative to all of these baselines.**
> >
> > We continue addressing this question in the next block.

---

> ### Author Response · Authors · 2025-11-21
> **Repsonse to Reviewer eaM8 (Part 4)**
>
> > Limited scope of evaluation. The experimental evaluation is narrow and lacks strong baselines. All experiments are conducted only on math-specific reasoning datasets (MATH500 and GSM-Plus), without testing on other reasoning or RLHF-related settings. Moreover, the paper compares CoLD only with simple internal variants and does not include key external baselines. Without such comparisons, it is difficult to assess whether CoLD offers any meaningful improvement over existing methods or if its gains merely result from additional regularization.
>
> - **Downstream RL application**. To more comprehensively demonstrate CoLD’s effectiveness on downstream tasks, we apply CoLD in a reinforcement learning setting. Specifically, our experiments are conducted within the ReasonFlux-PRM framework[6]. We use Qwen2.5-Math-1.5B-Instruct as the policy model and adopt GRPO (Group Relative Policy Optimization)[7] as the RL optimization algorithm. CoLD is integrated into the ReasonFlux-PRM and compared against its vanilla counterpart. For evaluation, we use MATH500 as the test benchmark.
> As shown in the table below, incorporating CoLD not only improves the accuracy of the policy model in answering questions, but also reduces the output length. This indicates that **CoLD effectively mitigates the length bias exhibited by the PRM, providing the model with more precise and meaningful reward signals during downstream reinforcement learning training**. Our RL experiments confirm that CoLD enhances PRM behavior not only at test time scaling but also during dynamic learning, where biased rewards would otherwise propagate and amplify over training steps.
>
> | Model                                             | Accuracy | Length |
> | ------------------------------------------------- | -------- | ------ |
> | Qwen2.5-Math-1.5B-Instruct+Vanilla ReasonFlux-PRM | 54.4     | 600.9  |
> | Qwen2.5-Math-1.5B-Instruct+CoLD ReasonFlux-PRM    | 57.6     | 543.2  |
>
>
> - **Cross-Domain Generalization**. While our primary experiments focus on mathematical reasoning, we have also evaluated CoLD in a broader setting by integrating it into VersaPRM [1], a process reward model designed to operate across multiple domains. We conduct our evaluation on the MMLU-Pro[2] benchmark, a more robust and challenging massive multi-task understanding dataset specifically designed to rigorously assess the capabilities of large language models. We applied CoLD in the domains of Computer Science, Economics, Chemistry, Biology, Philosophy and Other. Based on the results shown in the table above, when applying CoLD on top of VersaPRM, we observe consistent improvements in both accuracy and the reduction of response length across these heterogeneous domains. This demonstrates that the debiasing principle behind CoLD does not rely on domain-specific structures found in mathematical reasoning, but instead addresses a general phenomenon of superficial-feature bias in step-level reward modeling. **These results support the claim that CoLD generalizes beyond mathematics and can effectively enhance PRMs in diverse domain.**
>
>
>
> | Model            | Accuracy(Computer Science) | Length(Computer Science) | Accuracy(Economics) | Length(Economics) | Accuracy(Chemistry) | Length(Chemistry) |
> | ---------------- | -------------------------- | ------------------------ | ------------------- | ----------------- | ------------------- | ----------------- |
> | Vanilla VersaPRM | 56.3                       | 102.8                    | 47.8                | 102.8             | 56.4                | 88.3              |
> | CoLD VersaPRM    | 62.5                       | 90.8                     | 52.1                | 97.2              | 59.0                | 76.2              |
>
> | Model            | Accuracy(Biology) | Length(Biology) | Accuracy(Philosophy) | Length(Philosophy) | Accuracy(Other) | Length(Chemistry) |
> | ---------------- | ----------------- | --------------- | -------------------- | ------------------ | --------------- | ----------------- |
> | Vanilla VersaPRM | 73.6| 102.8    | 68.8  | 85.1   | 60.6            | 102.3             |
> | CoLD VersaPRM    | 78.9  | 90.8   | 68.8    | 80.2        | 66.7            | 91.9              |
>
> [1] Zeng T, Zhang S, Wu S, et al. VersaPRM: Multi-Domain Process Reward Model via Synthetic Reasoning Data
>
> [2] Wang Y, Ma X, Zhang G, et al. MMLU-Pro: A More Robust and Challenging Multi-Task Language Understanding Benchmark
>
> [3] Shen W, Zheng R, Zhan W, et al. Loose lips sink ships: Mitigating length bias in reinforcement learning from human feedback
>
> [4] Bu Y, Huo L, Jing Y, et al. Adaptive Length Bias Mitigation in Reward Models for RLHF
>
> [5] Huang Z, Qiu Z, Wang Z, et al. Post-hoc Reward Calibration: A Case Study on Length Bias
>
> [6] Zou J, Yang L, Gu J, et al. ReasonFlux-PRM: Trajectory-Aware PRMs forLong Chain-of-Thought Reasoning in LLMs
>
> [7] Shao Z, Wang P, Zhu Q, et al. DeepSeekMath: Pushing the Limits of Mathematical Reasoning in Open Language Models

---

> ### Author Response · Authors · 2025-11-27
>
> Dear Reviewer eaM8,
>
> I hope this message finds you well. We have carefully addressed the questions and concerns you previously raised, and we would greatly appreciate any further feedback you might be able to provide. Your insights are highly valuable to us, and we remain fully available to clarify or refine any remaining points.
>
> We sincerely appreciate your response at your earliest convenience, which will greatly assist us to move the discussion forward.
>
> Thank you sincerely for your time and effort in reviewing our paper.

---

> > ### Comment · Reviewer_eaM8 · 2025-11-28
> >
> > I appreciate the authors' additional experiments and clarifications. After reconsidering the overall contribution, I agree that the paper is not as weak as a '2' would suggest and I will update my score to 4.
> >
> > Regarding prior work (Weakness 2), although the authors demonstrate that existing length-bias methods perform less effectively in the PRM setting, my core concern remains: the proposed framework is still structurally very similar to earlier approaches (a learned bias component combined with a main reward model under joint training). CoLD is essentially an application of existing length-bias mitigation techniques to the PRM setting. (*I initially chose a 2 between 2 and 4 because the 4 rating explicitly states would not mind if the paper is accepted, as I still lean toward rejection.*)
> >
> > Therefore, the conceptual novelty feels marginal and the paper remains slightly below the acceptance threshold.

---

> > > ### Author Response · Authors · 2025-11-29
> > >
> > > Dear Reviewer eaM8,
> > >
> > > We sincerely appreciate your reconsideration and the updated score. We are glad that our additional experiments and clarifications have addressed your initial concerns.
> > >
> > > Regarding the conceptual novelty, we respectfully clarify that CoLD is not a mere application of existing techniques but **a framework specifically tailored for Process Reward Models (PRMs)**. We are the **first to identify and formalize the length-bias issue in this setting**, and since previous methods designed for Outcome Reward Models (ORMs) are ill-suited for step-wise supervision and yield poor performance, our approach offers a necessary and distinct solution that prior works cannot provide. We hope this underscores the unique value of our contribution.
> > >
> > > Thank you again for your constructive feedback and for helping us strengthen the paper.

---

### Official Review · Reviewer_aSou · 2025-10-28

**Soundness:** 2
**Presentation:** 4
**Contribution:** 2
**Rating:** 6
**Confidence:** 4

**Summary:**

This paper identifies and tackles a previously overlooked issue in Process Reward Models (PRMs) for mathematical reasoning—length bias, where longer reasoning steps receive higher rewards regardless of their semantic or logical validity. The authors design controlled experiments to empirically confirm this bias, then propose CoLD (Counterfactually-Guided Length Debiasing), a framework that mitigates length bias via causal analysis and counterfactual reasoning. CoLD integrates three complementary modules:
1. A length-penalty term to explicitly penalize verbosity.
2. A bias estimator that models spurious length-dependent effects.
3. Joint training between the PRM and estimator to disentangle correctness from surface length cues.

Extensive experiments on MATH500 and GSM-Plus datasets, across multiple policy and reward models (e.g., Llama-3-70B, MetaMath-Mistral-7B, Muggle-Math-13B), demonstrate that CoLD substantially reduces reward–length correlation and improves reasoning accuracy, producing shorter yet more correct reasoning trajectories.

**Strengths:**

1. Novel problem identification: The paper is the first to formally identify and empirically validate length bias in PRMs—a real and impactful issue that undermines reasoning evaluation reliability.

2. Causal-theoretic rigor: The causal graph formulation (S→L→P vs. S→C→P) clearly distinguishes spurious and genuine causal pathways, grounding the proposed framework in principled reasoning.

3. Systematic debiasing design: The three-module CoLD framework elegantly integrates heuristic (penalty), learnable (bias estimator), and structural (joint training) debiasing mechanisms.

4. Strong empirical results: Consistent improvements (+2–4% accuracy, −30% average length) across multiple datasets and models demonstrate robustness and practicality.

5. Practical significance: The framework can be applied post-hoc to existing PRMs without full retraining, which is valuable for real-world RLHF and inference-time alignment systems.

**Weaknesses:**

1. Limited theoretical novelty: While the combination of counterfactual reasoning and PRM debiasing is novel, each component (penalty, bias estimator, joint training) builds on existing ideas.

2. Lack of interpretability: The paper does not deeply analyze what the Bias Estimator learns or visualize its internal representations.

3. Domain specificity: Experiments are restricted to mathematical reasoning; it is unclear if CoLD generalizes to code reasoning, logical argumentation, or multimodal PRMs.

4. Missing behavioral analysis: The paper evaluates reward quality but not how CoLD impacts policy generation dynamics in RL training (e.g., entropy, verbosity collapse).

5. Presentation density: Some equations (Eq.3–9) are overly packed, and could be better supported with more intuitive diagrams or pseudo-code explanations.

**Questions:**

1. **Bias Estimator Interpretability**
   Could you provide more interpretability analysis of the *Bias Estimator*? For example, visualizing attention or gradient sensitivity to input length could clarify whether it truly captures spurious surface-level cues rather than semantic ones.

2. **Cross-domain Generalization**
   Have you examined whether **CoLD** generalizes beyond mathematical reasoning — e.g., to code reasoning or logical argumentation tasks? Such evidence would strengthen the generality claim.

3. **Counterfactual Sensitivity**
   In your counterfactual formulation (Eq. 10–13), have you tested sensitivity to the choice of counterfactual reference length \\( \ell^* \\)?

4. **Interaction with RL Fine-tuning**
   Could you provide insights on how **CoLD** interacts with reinforcement fine-tuning methods (e.g., PPO, GRPO)? Does debiasing reward signals change convergence speed or policy entropy?

5. **Potential Over-penalization**
   Are there any cases where **CoLD** over-penalizes necessary elaboration, resulting in shorter but incomplete reasoning?

**Details Of Ethics Concerns:**

This paper poses **no ethical risks** and requires **no ethics review**.

- The work uses only **public, non-sensitive mathematical reasoning datasets** (MATH500, GSM-Plus, PRM800K, Math-Shepherd).
- It does **not involve human subjects**, personal data, or privacy-related content.
- The proposed CoLD framework **reduces bias** in reward modeling rather than introducing new risks.
- All models and datasets used are under open licenses and comply with standard research-use terms.
- The contribution promotes **fairness, transparency, and responsible AI** practice.

**Conclusion:** No ethical concerns. The paper fully complies with the ICLR Code of Ethics.

---

> ### Author Response · Authors · 2025-11-21
> **Repsonse to Reviewer aSou (Part 1)**
>
> >Limited theoretical novelty: While the combination of counterfactual reasoning and PRM debiasing is novel, each component (penalty, bias estimator, joint training) builds on existing ideas.
>
> We thank the reviewer for raising the concern regarding theoretical novelty. We would like to clarify our contributions from several perspectives:
>
> 1. Although individual components of our method build on existing ideas, **our contribution lies in identifying and characterizing length bias in PRMs**. To our knowledge, no prior work has shown that PRMs consistently assign higher scores to longer steps even when the semantic correctness is unchanged.
>
> 2. **Using causal graph analysis, we show that step length acts as a confounding variable that interferes with reward estimation.** This perspective reveals why PRMs may behave unreliably in practice and highlights a structural issue that has not been previously analyzed.
>
> 3. Motivated by causal view, we propose the CoLD framework, which uses counterfactual reasoning to systematically reduce the influence of length. This **provides a principled and interpretable way to separate length-related effects from semantic reward signals**.
>
> 4. Across multiple tasks and datasets, **CoLD improves the stability and reliability of PRM-based task**, demonstrating that addressing length bias has both conceptual significance and practical utility beyond existing approaches.
>
> >Bias Estimator Interpretability:Could you provide more interpretability analysis of the Bias Estimator? For example, visualizing attention or gradient sensitivity to input length could clarify whether it truly captures spurious surface-level cues rather than semantic ones.
>
> We thank the reviewer for the valuable suggestion on improving the interpretability of the Bias Estimator. As a first step, we present a case study illustrating the Bias Estimator’s outputs.
>
> **Case Study:**
>
> **Question:** Convert the point $(0,3)$ in rectangular coordinates to polar coordinates.  Enter your answer in the form $(r,\theta),$ where $r > 0$ and $0 \le \theta < 2 \pi.$
>
> **Answer1:**
> > **Step 1.** Use the formulas $(r,\theta)$ satisfy $r = \sqrt{x^2 + y^2}, \theta = \operatorname{atan2}(y,x)$ **Step 2.** Compute $r$. Here $x=0$ and $y=3$, so $r = \sqrt{0^2 + 3^2} = \sqrt{9} = 3.$ **Step 3.** Determine $\theta$.  The point $(0,3)$ is on the positive $y$-axis, so $\theta = \frac{\pi}{2}$. **Step 4.** State the final answer:$(0,3) \longrightarrow (3, \tfrac{\pi}{2}).$
>
> Output of the **Bias Estimator** for each step:[0.2806,0.3257,0.3541,0.2556]
>
> **Answer2:**
> > **Step 1.** Use the formulas $(r,\theta)$ satisfy $r = \sqrt{x^2 + y^2}, \theta = \operatorname{atan2}(y,x)$ **Step 2.** Compute $r$. Here $x=0$ and $y=3$, so $r = \sqrt{0^2 + 3^2} = \sqrt{9} = 3.$ **Step 3.** Determine $\theta$.  Using $\theta = \arctan\left(\frac{y}{x}\right)$ gives $\theta = \arctan\left(\frac{3}{0}\right) = 0$. **Step 4.** State the final answer:$(0,3) \longrightarrow (3, 0)$.
>
> Output of the **Bias Estimator** for each step:[0.2806,0.3257,0.3491,0.2456]
>
> From this case study, we observe that the **Bias Estimator is driven more by length than by semantics.** For instance, in Step 3 of the two responses, despite differing semantic content and correctness, the Bias Estimator’s output is dominated by length.
>
> Furthermore, to illustrate this more clearly, we selected two sets of steps:
> **(A)** 30 steps that have identical length (average length of 20) but completely different meanings, and **(B)** 30 steps that differ in length but convey the same meaning. We then computed the variance of the Bias Estimator’s scores for these steps. The results show the following:
>
> | Identical Length with Different Meanings| Different Length with Identical Meanings |
> | -- | --|
> | 0.0115 | 0.4007 |
>
> The table shows that identical-length responses have smaller output variance, while varied-length responses show larger variance, further indicating that **the Bias Estimator is more sensitive to length and separates it from semantic content**.
>
> Additionally, to better examine whether the Bias Estimator relies on length rather than semantic information, we conduct two comparative attention analyses.
> - First, we feed Sentence A into both the Bias Estimator and the backbone model and compare their attention distributions. While the backbone exhibits semantically focused attention peaks, the Bias Estimator assigns attention almost uniformly across tokens, without concentrating on semantic-bearing words.
> - Second, we input Sentence A and Sentence B, which share the same length but differ completely in semantics, into the Bias Estimator. Their attention maps remain nearly indistinguishable, indicating that the attention pattern remains stable when semantics change while length is kept constant.
>
> **These results confirm that the Bias Estimator primarily encodes length rather than semantic cues.** Due to OpenReview limitations, figures are provided instead in Appendix H.2.

---

> ### Author Response · Authors · 2025-11-21
> **Repsonse to Reviewer aSou (Part 2)**
>
> >Cross-domain Generalization: Have you examined whether CoLD generalizes beyond mathematical reasoning — e.g., to code reasoning or logical argumentation tasks? Such evidence would strengthen the generality claim.
>
> We thank the reviewer for this insightful question regarding cross-domain generalization. While our primary experiments focus on mathematical reasoning, we have also evaluated CoLD in a broader setting by integrating it into VersaPRM [1], a process reward model designed to operate across multiple domains. We conduct our evaluation on the MMLU-Pro[2] benchmark, a more robust and challenging massive multi-task understanding dataset specifically designed to rigorously assess the capabilities of large language models. We applied CoLD in the domains of Computer Science, Economics, Chemistry, Biology, Philosophy, Other and the results are presented below:
>
> | Model            | Accuracy(Computer Science) | Length(Computer Science) | Accuracy(Economics) | Length(Economics) | Accuracy(Chemistry) | Length(Chemistry) |
> | ---------------- | -------------------------- | ------------------------ | ------------------- | ----------------- | ------------------- | ----------------- |
> | Vanilla VersaPRM | 56.3                       | 102.8                    | 47.8                | 102.8             | 56.4                | 88.3              |
> | CoLD VersaPRM    | 62.5                       | 90.8                     | 52.1                | 97.2              | 59.0                | 76.2              |
>
> | Model            | Accuracy(Biology) | Length(Biology) | Accuracy(Philosophy) | Length(Philosophy) | Accuracy(Other) | Length(Chemistry) |
> | ---------------- | ----------------- | --------------- | -------------------- | ------------------ | --------------- | ----------------- |
> | Vanilla VersaPRM | 73.6              | 102.8           | 68.8                 | 85.1               | 60.6            | 102.3             |
> | CoLD VersaPRM    | 78.9              | 90.8            | 68.8                 | 80.2               | 66.7            | 91.9              |
>
> Based on the results shown in the table above, when applying CoLD on top of VersaPRM, we observe consistent improvements in both accuracy and the reduction of response length across these heterogeneous domains. This demonstrates that the debiasing principle behind CoLD does not rely on domain-specific structures found in mathematical reasoning, but instead addresses a general phenomenon of superficial-feature bias in step-level reward modeling. **These results support the claim that CoLD generalizes beyond mathematics and can effectively enhance PRMs in diverse domain.**
>
> [1] Zeng T, Zhang S, Wu S, et al. VersaPRM: Multi-Domain Process Reward Model via Synthetic Reasoning Data
>
> [2] Wang Y, Ma X, Zhang G, et al.  MMLU-Pro: A More Robust and Challenging Multi-Task Language Understanding Benchmark
>
> >Counterfactual Sensitivity:In your counterfactual formulation (Eq. 10–13), have you tested sensitivity to the choice of counterfactual reference length $l^\star$
>
> We sincerely thank the reviewer for raising this question and appreciate the opportunity to clarify the causal role of the counterfactual reference length $l^\star$, which may have led to some misunderstanding.
>
> In our counterfactual formulation (Eq. 10–13), **$l^\star$ is no a tunable hyperparameter and therefore does not require sensitivity analysis.** Instead, $l^*$ is the counterfactual intervention value used to isolate the causal contribution of length to the PRM score. Formally, when computing
> $$
> E_{C,N \rightarrow P} = P(c,n,\ell) - P(c^{\star}, n^{\star}, \ell),
> $$
> the term $P(c^{\star}, n^{\star}, l^{\star})$ is only an intermediate quantity that helps decompose the total effect into the contribution from length $(E_{L\rightarrow P})$ and the contribution from semantic factors $(E_{C,N\rightarrow P})$. After simplification, the resulting causal effect on the prediction depends only on $\ell$, the actual observed length, and not on $l^\star$. In other words, $l^\star$ cancels out analytically:
>
> $$
> E_{C,N\rightarrow P}
> = P(c,n,\ell)-P(c^{\star}, n^{\star}, \ell),
> $$
>
> which **means that the final counterfactual estimate is invariant to the choice of $l^\star$.** This property is exactly why the decomposition in Eq. 10–13 is constructed in this way—it ensures that the causal effect of length can be cleanly removed without introducing sensitivity to an arbitrary reference value.
>
> Therefore, testing sensitivity with respect to $l^\star$ is unnecessary, since $l^\star$ is not a free design choice and does not influence the causal effect that CoLD ultimately computes.

---

> > ### Comment · Reviewer_aSou · 2025-11-24
> > **So in your design, is the length penalty a manually specified parameter?**
> >
> > So in your design, is the length penalty a manually specified parameter?

---

> ### Author Response · Authors · 2025-11-21
> **Repsonse to Reviewer aSou (Part 3)**
>
> >Interaction with RL Fine-tuning: Could you provide insights on how CoLD interacts with reinforcement fine-tuning methods (e.g., PPO, GRPO)? Does debiasing reward signals change convergence speed or policy entropy?
>
> We thank the reviewer for this insightful question regarding the interaction between CoLD and reinforcement learning fine-tuning methods. In recent research, many works [1][2][3][4] have adopted PRMs to provide step-level reward signals for reinforcement learning, where the PRM plays a central role in guiding the policy model throughout training. However, **the effectiveness of such RL pipelines relies on the PRM’s ability to deliver reward signals that truly reflect correctness.** If the PRM’s scores are influenced by superficial features such as response length rather than genuine semantic accuracy, **these biases can undermine the reliability of the reward signal.**
>
> For instance, if a PRM tends to assign higher scores to longer responses, the policy model may learn to produce unnecessarily long outputs simply because they receive higher rewards, even when the content itself is not more correct. This can lead to reward hacking, where the model optimizes for artifacts instead of the intended objective.
>
> To more comprehensively demonstrate CoLD’s effectiveness on downstream tasks, **we further apply CoLD in a reinforcement learning setting**. Specifically, our experiments are conducted within the ReasonFlux-PRM framework[4]. We use Qwen2.5-Math-1.5B-Instruct as the policy model and adopt GRPO(Group Relative Policy Optimization)[5] as the RL optimization algorithm. CoLD is integrated into the ReasonFlux-PRM and compared against its vanilla counterpart. For evaluation, we use MATH500 as the test benchmark. The results are summarized below.
>
>
> | Model                                             | Accuracy | Length |
> | ------------------------------------------------- | -------- | ------ |
> | Qwen2.5-Math-1.5B-Instruct+Vanilla ReasonFlux-PRM | 54.4     | 600.9  |
> | Qwen2.5-Math-1.5B-Instruct+CoLD ReasonFlux-PRM    | 57.6     | 543.2  |
>
> As shown in the table above, incorporating CoLD not only improves the accuracy of the policy model in answering questions, but also reduces the output length. **This indicates that CoLD effectively mitigates the length bias exhibited by the PRM, providing the model with more precise and meaningful reward signals during downstream reinforcement learning training.**
>
> [1] Wang P, Li L, Shao Z, et al. Math-Shepherd: Verify and Reinforce LLMs Step-by-step without Human Annotations
>
> [2] Wang J, Fang M, Wan Z, et al. OpenR: An Open Source Framework for Advanced Reasoning with Large Language Models
>
> [3] Xie G, Shi Y, Tian H, et al. CAPO: Towards Enhancing LLM Reasoning through Verifiable Generative Credit Assignment
>
> [4] Zou J, Yang L, Gu J, et al. ReasonFlux-PRM: Trajectory-Aware PRMs forLong Chain-of-Thought Reasoning in LLMs
>
> [5] Shao Z, Wang P, Zhu Q, et al. DeepSeekMath: Pushing the Limits of Mathematical Reasoning in Open Language Models
>
> > Presentation density: Some equations (Eq.3–9) are overly packed, and could be better supported with more intuitive diagrams or pseudo-code explanations.
>
> We appreciate the reviewer’s feedback on presentation clarity. To improve accessibility, our paper already includes multiple supporting materials designed to unpack the equations.
> - The right panel of Figure 3 provides **a concise flowchart that visually demonstrates how Eq. 3 is computed in practice**, making the procedural steps more intuitive.
> - Appendix B offers **an extended algorithm description together with detailed pseudocode** (Algorithm 1: Joint Training of PRM and Bias Estimator), which explicitly breaks down how Eqs. 3–9 interact during training.
>
> These additions were intended to help readers connect the mathematical formulation with the actual implementation workflow. If there are still parts of the presentation that feel unclear or if any step of the algorithm remains ambiguous, we would be grateful for further suggestions.

---

> > ### Author Response · Authors · 2025-11-21
> > **Repsonse to Reviewer aSou (Part 4)**
> >
> > > Potential Over-penalization: Are there any cases where CoLD over-penalizes necessary elaboration, resulting in shorter but incomplete reasoning?
> >
> > We thank the reviewer for raising this concern. CoLD is designed to mitigate length bias without discouraging necessary elaboration. The main factors affecting the penalty strength are two parameters, $c$ and $\alpha$. In Appendix E.1, we systematically study how different choices of $c$ impact overall performance. The experiments show that **CoLD exhibits strong robustness to the choice of $c$, and no significant over-penalization is observed**.
> >
> > To further analyze potential over-penalization, we conduct additional experiments on the parameter $\alpha$ across three Policy Model. The specific results are shown in the table below:
> >
> >
> > **Policy Model:llama3-70b-instruct**
> > | Alpha   | ArithACC(MATH500) | Length(MATH500) | ArithACC(GSM-Plus) | Length(GSM-Plus) |
> > | ------- | ----------------- | --------------- | ------------------ | ---------------- |
> > | 0.1     | 38.8              | 222.6           | 68.6               | 177.4            |
> > | 0.01    | 42.4              | 223.5           | 71.1               | 179.6            |
> > | 0.001   | 48.6              | 253.5           | 72.6               | 189.7            |
> > | 0.0001  | 49.2              | 313.2           | 73.8               | 202.5            |
> > | 0.00001 | 47.9              | 350.6           | 73.3               | 210.4            |
> >
> > **Policy Model:Muggle-Math-13B**
> > | Alpha   | ArithACC(MATH500) | Length(MATH500) | ArithACC(GSM-Plus) | Length(GSM-Plus) |
> > | ------- | ----------------- | --------------- | ------------------ | ---------------- |
> > | 0.1     | 21.4              | 191.1           | 43.7               | 193.1            |
> > | 0.01    | 27.0              | 195.9           | 50.0               | 195.8            |
> > | 0.001   | 30.2              | 248.4           | 58.5               | 215.8            |
> > | 0.0001  | 31.4              | 309.2           | 60.3               | 243.3            |
> > | 0.00001 | 30.2              | 357.7           | 59.62              | 263.2            |
> >
> > **Policy Model:MetaMath-Mistral-7B**
> > | Alpha   | ArithACC(MATH500) | Length(MATH500) | ArithACC(GSM-Plus) | Length(GSM-Plus) |
> > | ------- | ----------------- | --------------- | ------------------ | ---------------- |
> > | 0.1     | 25.0              | 215.3           | 45.5               | 202.9            |
> > | 0.01    | 31.2              | 218.9           | 50.2               | 205.3            |
> > | 0.001   | 38.8              | 276.29          | 56.9               | 220.2            |
> > | 0.0001  | 37.2              | 376.3           | 61.4               | 238.6            |
> > | 0.00001 | 36.2              | 393.8           | 59.9               | 259.4            |
> >
> >
> > From the table, we can see that within a reasonable range of $\alpha$, CoLD demonstrates robustness, and the strength of the penalty controlled by $\alpha$ has very limited impact on performance. However, it is also clear that when $\alpha$ applies an excessively strong penalty, over-penalization occurs. This indicates that **over-penalization can appear in certain extreme cases.** Nevertheless, **as long as $\alpha$ or $c$ is chosen within a reasonable range, CoLD exhibits strong robustness and avoids over-penalization** without the need for very fine-grained tuning.

---

> ### Author Response · Authors · 2025-11-24
>
> Thank you for this insightful question. The short answer is **yes**, the coefficient $\alpha$ in the explicit **Length Penalty** term ($-\alpha \ell(x)$) is a manually specified hyperparameter, as described in Section 3.2.
>
> While the explicit length penalty ($\alpha$) provides a direct constraint against verbosity, the contribution of CoLD also lies in the **Trainable Bias Estimator**. Unlike a fixed manual parameter, the Bias Estimator is dynamically trained to capture non-linear, spurious correlations between length and reward, separating them from true reasoning quality based on our causal graph analysis (Section 2.2).
>
> **Why Manual Specification is Not Enough:**
>
> Relying solely on a manually specified $\alpha$ implies that the length bias is strictly linear and uniform across all samples. This is a heuristic that fails to capture complex dependencies. By introducing the **Bias Estimator**, CoLD dynamically learns the specific bias component for each reasoning step, allowing for a more granular **counterfactual adjustment** ($r^*(x)$ in Eq. 3) that subtracts the estimated spurious signal.
>
>
> In summary, $\alpha$ serves as a heuristic regularizer, while the learnable Bias Estimator performs the core causal intervention required to robustly mitigate length bias.

---

### Official Review · Reviewer_MDNX · 2025-10-31

**Soundness:** 2
**Presentation:** 1
**Contribution:** 1
**Rating:** 2
**Confidence:** 3

**Summary:**

To tackle the verbosity bias of process reward models, this paper proposes to postprocess PRM predictions with lengh penalty and learnt bias estimator.

**Strengths:**

1. The motivation is clear.
2. The experiment result supports the paper's claim.

**Weaknesses:**

1. Unclear and insufficient discussion on implementation and design choice. See questions below.
2. Verbosity bias have been observed long time ago in outcome-based reward models and LLM judges. Although this paper tackles PRM, there seems no specific design utilizing the special nature of process modeling.

**Questions:**

1. Implementation:
- How to add Gaussian noise on tokens (sec 3.3)?
- How exactly do gradient pass through Pearson correlation (eq 4, 5)? Is anything (e.g. inner expectation, standard deviations) detached?
- There are so many hyperparameters (e.g. c, alpha in eq 3, lambda  n eq 6, 7). How to decide them in practice?
2. Design:
- Eq 3 is a complex heuristic. Why choose such a complex design? Can simpler objectives (e.g. a straightforward $r_\theta(x) + b_\phi(x)$) do the trick?
- Why don't train eq 3 end-to-end?
- Optimizing the bias estimator with Pearson correlation violates Goodhart's law (When a measure becomes a target, it ceases to be a good measure). Is there any comment on this?

---

> ### Author Response · Authors · 2025-11-21
> **Repsonse to Reviewer MDNX (Part 1)**
>
> > Verbosity bias have been observed long time ago in outcome-based reward models and LLM judges. Although this paper tackles PRM, there seems no specific design utilizing the special nature of process modeling.
>
> We thank the reviewer for this comment. While prior work has largely focused on outcome-level scoring, **our work specifically addresses PRMs, which operate on step-level reasoning.** To clarify the unique challenges faced by PRMs compared with outcome-based reward models, we highlight several distinctions below:
> 1. **PRMs are inherently more sensitive than outcome-based reward models (ORMs)** because they evaluate the model’s reasoning at each intermediate step rather than only the final answer. To illustrate this, we analyzed the variance of step-level lengths and response-level lengths on the PRM800K[4] dataset. The results, summarized in the table below, show that step-level lengths exhibit higher variance than overall response lengths. This indicates that scoring at the step level is a more complex task and is more susceptible to superficial biases, such as verbosity, and to potential reward hacking.
>
>
> |                | SD    |
> | -------------- | ----- |
> | step-level     | 71.7  |
> | response-level | 50.6  |
>
>
> 2. **We compared several debiasing methods originally designed for outcome-based reward models and LLM judges to demonstrate why such designs may not transfer well to the PRM setting**. In the table, LLSS refers to Loose Lips Sink Ships [1], ALBM to Adaptive Length Bias Mitigation [2], UA to Uniform Average, and LWR to Locally Weighted Regression [3]. LWR was already reported as a baseline in Table 1 and is repeated here for ease of direct comparison.
>
>     From the experimental results, we observe that although **these methods effectively address length bias in traditional reward modeling, they perform noticeably worse in the PRM setting**, exhibiting subpar results in both accuracy and the length of the selected responses. In contrast, our CoLD method successfully mitigates the length bias that emerges in PRM.
>
> **Policy Model:llama3-70b-instruct**
> | Method     | ArithACC(MATH500) | Length(MATH500) | ArithACC(GSM-Plus) | Length(GSM-Plus) |
> | ---------- | ----------------- | --------------- | ------------------ | ---------------- |
> | LLSS       | 45.8              | 435.7           | 72.5               | 302.5            |
> | ALBM       | 43.0              | 594.8           | 72.2               | 364.3            |
> | UA         | 40.4              | 608.5           | 70.4               | 324.6            |
> | LWR        | 39.0              | 590.7           | 69.6               | 299.9            |
> | CoLD(Ours) | 49.2              | 313.2           | 73.8               | 202.5            |
>
> **Policy Model:Muggle-Math-13B**
> | Method| ArithACC(MATH500) | Length(MATH500) | ArithACC(GSM-Plus) | Length(GSM-Plus) |
> | ---------- | ----------------- | --------------- | ------------------ | ---------------- |
> | LLSS       | 28.6              | 375.7           | 59.1               | 263.9            |
> | ALBM       | 25.4              | 472.0           | 56.3               | 364.5            |
> | UA         | 27.4              | 440.8           | 55.3               | 328.7            |
> | LWR        | 23.5              | 470.5           | 50.7               | 312.3            |
> | CoLD(Ours) | 31.4              | 309.2           | 60.3               | 243.3            |
>
> **Policy Model:MetaMath-Mistral-7B**
> | Method| ArithACC(MATH500) | Length(MATH500) | ArithACC(GSM-Plus) | Length(GSM-Plus) |
> | ---------- | ----------------- | --------------- | ------------------ | ---------------- |
> | LLSS | 34.5     | 441.2  | 58.0               | 340.0            |
> | ALBM | 31.0      | 610.0     | 56.5               | 385.0            |
> | UA | 33.2 | 609.9           | 56.3               | 363.8            |
> | LWR | 29.6    | 507.3           | 52.7               | 354.4            |
> | CoLD(Ours) | 37.2  | 376.3           | 61.4               | 238.6            |
>
> 3. **CoLD is designed to leverage this process-level nature**: by disentangling superficial features such as length from semantic correctness in each reasoning step, it corrects biases that propagate through the entire reasoning trajectory. This is distinct from outcome-level debiasing, because PRMs evaluate the reasoning as a sequence of steps, not just the final answer. Our method explicitly exploits this property by modeling and adjusting for length bias at the step level, ensuring that the bias-only signal does not interfere with the stepwise learning of semantic correctness.
>
> [1]Shen W, Zheng R, Zhan W, et al. Loose lips sink ships: Mitigating length bias in reinforcement learning from human feedback
>
> [2]Bu Y, Huo L, Jing Y, et al. Adaptive Length Bias Mitigation in Reward Models for RLHF
>
> [3]Huang Z, Qiu Z, Wang Z, et al. Post-hoc Reward Calibration: A Case Study on Length Bias
>
> [4]Lightman H, Kosaraju V, Burda Y, et al. Let's verify step by step

---

> > ### Author Response · Authors · 2025-11-21
> > **Repsonse to Reviewer MDNX (Part 2)**
> >
> > >How to add Gaussian noise on tokens (sec 3.3)?
> >
> > We thank the reviewer for the question regarding how Gaussian noise is added to tokens in Section 3.3. To ensure that the bias-only model effectively captures the length bias present in the input, we intentionally inject random noise into the token representations[1]. This noise disrupts the semantic content of each token, helping the model focus on superficial features such as length rather than the underlying meaning. Formally, **for each input token id $x$, we generate a perturbed token value as $\tilde{x}$**
> >
> > $$\tilde{x} = \text{clip}\bigl(x + 5\epsilon,; 0,; V-1\bigr), \quad \epsilon \sim \mathcal{N}(0,1),; \epsilon \in [-1,1]$$
> >
> > The noise term $\epsilon$ is drawn from a standard Gaussian distribution truncated to the interval $[-1, 1]$. For each input token id $x$, a bounded random perturbation is added, and the result is clipped to the valid vocabulary range $[0, V-1]$. Finally, the values are rounded down to obtain the new discrete token sequence.
> >
> > [1]Loose lips sink ships: Mitigating Length Bias in Reinforcement Learning from Human Feedback
> >
> > > How exactly do gradient pass through Pearson correlation (eq 4, 5)? Is anything (e.g. inner expectation, standard deviations) detached?
> >
> > We thank the reviewer for the insightful question. To clarify, **the gradient passes through all components of the correlation computation, and no part of the inner expectation or the standard deviations is detached.** To make this explicit, we first derive the formula $\frac{\partial \rho}{\partial r_k}$.
> >
> > Denote $r_i = r_\theta(x_i)$.
> > The sample mean, covariance, and standard deviations are defined as
> >
> > $$
> > \bar r=\frac{1}{N}\sum_{i=1}^N r_i,\qquad
> > \bar\ell=\frac{1}{N}\sum_{i=1}^N \ell_i,
> > $$
> >
> > $$
> > \mathrm{Cov}=\frac{1}{N}\sum_{i=1}^N (r_i-\bar r)(\ell_i-\bar\ell),\qquad
> > \sigma_r=\sqrt{\frac{1}{N}\sum_{i=1}^N (r_i-\bar r)^2},\qquad
> > \sigma_\ell=\sqrt{\frac{1}{N}\sum_{i=1}^N(\ell_i-\bar\ell)^2}.
> > $$
> >
> > The sample Pearson correlation is
> >
> > $$
> > \rho=\frac{\mathrm{Cov}}{\sigma_r\sigma_\ell}.
> > $$
> >
> > We differentiate $\rho$ with respect to a single entry $r_k$. First, the derivative of the sample mean is
> >
> > $$
> > \frac{\partial \bar r}{\partial r_k}=\frac{1}{N}.
> > $$
> >
> > Using the definition of covariance and the identity
> > $$
> > \frac{\partial (r_i-\bar r)}{\partial r_k}=\delta_{ik}-\frac{1}{N} ,
> > $$
> > (where $\delta_{ik}$ is the Kronecker delta), we compute
> > $$
> > \frac{\partial \mathrm{Cov}}{\partial r_k}
> > =\frac{\partial}{\partial r_k}\Big(\frac{1}{N}\sum_{i=1}^N (r_i-\bar r)(\ell_i-\bar\ell)\Big)
> > =\frac{1}{N}\sum_{i=1}^N (\delta_{ik}-\tfrac{1}{N})(\ell_i-\bar\ell).
> > $$
> > Since $\sum_{i=1}^N(\ell_i-\bar\ell)=0$, this simplifies to
> > $$
> > \frac{\partial \mathrm{Cov}}{\partial r_k}=\frac{1}{N}(\ell_k-\bar\ell).
> > $$
> > Next, we differentiate the standard deviation. Start from
> > $$
> > \sigma_r^2=\frac{1}{N}\sum_{i=1}^N (r_i-\bar r)^2.
> > $$
> > Then
> > $$
> > \frac{\partial \sigma_r^2}{\partial r_k}
> > =\frac{1}{N}\sum_{i=1}^N 2(r_i-\bar r)(\delta_{ik}-\tfrac{1}{N})
> > =\frac{2}{N}(r_k-\bar r),
> > $$
> > which yields
> > $$
> > \frac{\partial \sigma_r}{\partial r_k}
> > =\frac{1}{2\sigma_r}\frac{\partial \sigma_r^2}{\partial r_k}
> > =\frac{1}{N\sigma_r}(r_k-\bar r).
> > $$
> >
> > Now differentiate $\rho=\dfrac{\mathrm{Cov}}{\sigma_r\sigma_\ell}$. Since $\sigma_\ell$ does not depend on $r_k$,
> > $$
> > \frac{\partial \rho}{\partial r_k}
> > =\frac{1}{\sigma_\ell}\frac{\partial}{\partial r_k}\Big(\frac{\mathrm{Cov}}{\sigma_r}\Big)
> > =\frac{1}{\sigma_\ell}\cdot\frac{\sigma_r\frac{\partial\mathrm{Cov}}{\partial r_k}-\mathrm{Cov}\frac{\partial\sigma_r}{\partial r_k}}{\sigma_r^2}.
> > $$
> > Substituting the two derivatives above and simplifying gives
> > $$
> > \frac{\partial \rho}{\partial r_k}
> > =\frac{1}{N}\left(\frac{\ell_k-\bar\ell}{\sigma_r\sigma_\ell}-\frac{\mathrm{Cov},(r_k-\bar r)}{\sigma_r^3\sigma_\ell}\right).
> > $$
> >
> > Thus, the continuous derivation and final sample-based expression for the derivative of the correlation with respect to the individual model output $r_k = r_\theta(x_k)$ is
> > $$
> > \frac{\partial \rho}{\partial r_k}
> > =\frac{1}{N}\frac{\ell_k-\bar\ell}{\sigma_r\sigma_\ell}
> > -\frac{1}{N}\frac{\mathrm{Cov},(r_k-\bar r)}{\sigma_r^3\sigma_\ell}
> > $$
> >
> > Then
> > $$\frac{\partial \mathcal{L}_{\text{PRM}}({\theta})}{\partial \theta}=2\lambda_r \rho_r \frac{\partial \rho_r}{\partial \theta}$$
> >
> > $$\frac{\partial \rho_r}{\partial \theta}=\frac{\partial r_\theta}{\partial \theta}\frac{\partial \rho_r}{\partial r_\theta}$$
> >
> > The derivation of $\frac{\partial \mathcal{L}_{\text{bias}}({\phi})}{\partial \phi}$ is similar as well.
> >
> > **In our implementation, the mean, covariance, and standard deviation are all fully differentiable and kept undetached, allowing gradients to propagate completely through these computations back to the model parameters.** In addition, we include a small constant $\varepsilon$ to prevent division by zero, as shown in the following formula:
> >
> >
> > $$\sigma_r=\sqrt{\frac{1}{N}\sum_{i=1}^N (r_i-\bar r)^2+\varepsilon},\qquad
> > \sigma_\ell=\sqrt{\frac{1}{N}\sum_{i=1}^N(\ell_i-\bar\ell)^2+\varepsilon}.$$

---

> > > ### Author Response · Authors · 2025-11-21
> > > **Repsonse to Reviewer MDNX (Part 3)**
> > >
> > > >There are so many hyperparameters (e.g. c, alpha in eq 3, lambda n eq 6, 7). How to decide them in practice?
> > >
> > > In our experiments, **we systematically explored the ranges of key hyperparameters using grid search**. The hyperparameter study presented in Appendix E shows that **our method is highly robust to these choices**. Even when the selected hyperparameter combination deviates slightly from the optimal one, the model still performs well on the main evaluation metrics.
> > >
> > > > Eq 3 is a complex heuristic. Why choose such a complex design? Can simpler objectives (e.g. a straightforward $r_\theta(x)+b_\phi(x)$) do the trick?
> > >
> > > >Why don't train eq 3 end-to-end?
> > >
> > > Thank you for the thoughtful question. Although Eq. 3 may look complex at first glance, its structure is not a heuristic choice—we derive it directly from the counterfactual causal decomposition in our causal graph. Below we explain (1) why Eq. 3 must take this form, (2) why simpler formulations fail, and (3) why Eq. 3 cannot be trained end-to-end.
> > >
> > >
> > > **(1) why the structure is necessary**
> > >
> > > The expression in Eq. 3 corresponds exactly to the counterfactual effect (A detailed analysis can be found in Appendix A)
> > > $$
> > > E_{C,N \rightarrow P} = P(C=c,N=n,L=\ell) - P(C=c^\star,N=n^\star,L=\ell),
> > > $$
> > > which isolates the causal contribution of correctness while holding the length fixed. **Each term in Eq. 3 has a direct semantic meaning within this decomposition**: the composed reward $r_\theta(x)\sigma(b_\phi(x+\mathcal{N}))$ estimates the factual quantity $P(C=c,N=n,L=\ell)$, while $\sigma(b_\phi(x+\mathcal{N}))$ and $\alpha \ell(x)$ serve as approximations of the counterfactual baseline $P(C=c^\star,N=n^\star,L=\ell)$. The structure thus follows from the algebra of counterfactual inference rather than heuristic trial-and-error.
> > >
> > > The apparent "complexity" is therefore a reflection of the causal structure we aim to model. The correctness effect cannot be isolated using a single function of $x$ because $P$ depends jointly on both $C,N$ and $L$. Eq. 3 is simply the minimal operational form needed to compute the factual–counterfactual difference that defines the causal effect.
> > >
> > > **(2) why simpler formulations fail**
> > >
> > > A simple expression such as $r_\theta(x)+b_\phi(x)$ implicitly assumes that correctness and length independently influence the prediction. This would correspond to a factorization $P(C,L) = P(C)+P(L)$, **which contradicts the causal graph where both variables interact in determining the PRM output.** As a result, additive formulations cannot disentangle correctness from length; they treat the length-induced bias as just another additive signal rather than a confounder that must be explicitly removed.
> > >
> > > Moreover, debiasing requires measuring the difference between factual and counterfactual predictions. A simple additive correction only approximates the factual value $P(C=c,L=\ell)$, without providing any mechanism to estimate the counterfactual baseline where correctness is removed but length is preserved. Without this baseline, the model cannot isolate the causal correctness effect and inevitably continues to inherit length bias.
> > >
> > > To better address your concerns, **We also experimentally evaluated several simplified designs**, including $r_\theta(x)+b_\phi(x)$, $r_\theta(x)-b_\phi(x)$. The results are summarized in the table below, where we used Llama-3-70B-Instruct as the policy model.
> > >
> > > | Equation | ArithACC(MATH500) | Length(MATH500) | ArithACC(GSM-Plus) | Length(GSM-Plus) |
> > > | --- | --- | --- |--| -- |
> > > | $r_\theta(x)+b_\phi(x)$| 39.8 | 574.9 | 67.2  | 332.5 |
> > > | $r_\theta(x)-b_\phi(x)$| 42.6 | 545.6  | 71.8 | 301.6 |
> > > | CoLD(Ours)-w/o Penalty | 48.2 | 494.9 | 73.2| 237.4|
> > >
> > > As shown in the table, these simpler objectives exhibit relatively modest performance.
> > >
> > > **(3) why Eq. 3 cannot be trained end-to-end**
> > >
> > > Training Eq. 3 directly would require supervision for both the factual quantity $P(C=c,N=n,L=\ell)$ and the counterfactual baseline $P(C=c^\star,N=n^\star,L=\ell)$. However, the latter does not appear in any PRM dataset, because we never observe a version of the same step where correctness is removed while length is held unchanged. This lack of counterfactual labels is precisely why causal inference is needed in the first place.
> > >
> > > **Because the counterfactual portion of Eq. 3 is not directly observable, end-to-end training would be underdetermined.** The model could satisfy the training objective through arbitrary internal representations that do not correspond to the intended causal decomposition, leading to degenerate solutions. To prevent this, **we train each component only where supervision exists and perform the factual–counterfactual subtraction at inference time**, when both estimates are available.
> > >
> > > This two-stage procedure is thus not an artifact of design but a necessity imposed by identifiability. Without separate estimation of factual and counterfactual components, the model cannot recover the correctness effect or reliably remove length bias.

---

> > > > ### Author Response · Authors · 2025-11-21
> > > > **Repsonse to Reviewer MDNX (Part 4)**
> > > >
> > > > >optimizing the bias estimator with Pearson correlation violates Goodhart's law (When a measure becomes a target, it ceases to be a good measure). Is there any comment on this?
> > > >
> > > > Thank you for raising this insightful point regarding Goodhart’s law. We would like to clarify why optimizing the bias estimator using Pearson correlation does not lead to the measure–target collapse implied by Goodhart’s law:
> > > >
> > > > 1. The Pearson correlation objective is **not aligned with any final evaluation metric**. Neither accuracy nor response length enters the training objective; **length reduction appears only as a byproduct of mitigating superficial-feature bias**, not as something being optimized for.
> > > >
> > > > 2. The metric optimized during training (correlation between length and PRM scores) is **structurally different from the metrics used at test time**. Pearson correlation is used solely to train the internal bias estimator on step-level annotations, whereas evaluation involves Best-of-n(Test-time Scaling), ProcessBench results, and downstream RL tasks. Since none of these tasks optimize Pearson correlation directly, there is no feedback loop that would cause the measure to cease being meaningful.
> > > >
> > > > 3. The deployment setting creates a natural insulation from Goodhart-style failures. The bias estimator is **trained under a step-wise scoring formulation but is applied only indirectly—through debiasing the PRM used for reranking or guiding Test-time Scaling/RL.** These downstream objectives never target the correlation metric or length itself.

---

> ### Author Response · Authors · 2025-11-27
>
> Dear Reviewer MDNX,
>
> I hope this message finds you well. We have carefully addressed the questions and concerns you previously raised, and we would greatly appreciate any further feedback you might be able to provide. Your insights are highly valuable to us, and we remain fully available to clarify or refine any remaining points.
>
> We sincerely appreciate your response at your earliest convenience, which will greatly assist us to move the discussion forward.
>
> Thank you sincerely for your time and effort in reviewing our paper.

---

### Author Response · Authors · 2025-11-21

We sincerely thank the Program Chairs, the Senior Area Chairs, the Area Chairs and all the reviewers for their hard work and dedication in evaluating our paper.


We have updated the PDF version of the paper with the following changes, hoping that these improvements help clarify our contributions and address the reviewers’ concerns.:

1. Added two **additional baseline** comparisons in Table 1.
2. Added Section 4.6, *Downstream RL Application*, including experiments applying CoLD to **downstream RL** tasks.
3. Added Section 4.7, *Cross-Domain Generalization Study*, including experiments applying CoLD **across different domains**.
4. Revised the description in Section 4.8 and added a brief introduction to the corresponding experiments.
5. Expanded Appendix C.1, *Implementation Details*, with descriptions of the newly added baselines and base models.
6. Added Appendix E.1, *Effect of $\alpha$*, which includes an experimental analysis of the **$\alpha$ parameter**.
7. Added Appendix G, *Evaluation Using Stronger Policy Model*, including experiments based on **two stronger policy models**.
8. Added Appendix H, *Bias Estimator Interpretability*, which contains experiments and analyses on **the interpretability of the Bias Estimator**.

We use blue to highlight the corresponding changes.

---

### Author Response · Authors · 2025-12-01
**Summary of Revisions and Clarifications**

Dear AC,

We sincerely thank the Area Chair and all reviewers for their constructive feedback. Below, we summarize our responses and the corresponding revisions, clustered by key topics:

**1. Comprehensive Evaluation & Baseline Comparison**

*  **[R2, R3, R4]Downstream RL:** We integrated CoLD into a reinforcement learning pipeline (GRPO), confirming that it effectively guides policy optimization and mitigates reward hacking in dynamic training.
*   **[R2, R3]Cross-Domain Generalization:** Verified CoLD on the MMLU-Pro benchmark (CS, Biology, etc.), showing effectiveness beyond mathematical reasoning.
*   **[R4]Strong Policy Models:** Validated CoLD on Qwen3-Next-80B and DeepSeek-v3.1, proving consistent improvements on state-of-the-art large-scale models.
*   **[R1, R3]Stronger Baselines:** We added comparisons with outcome-based debiasing methods (e.g., LLSS, ALBM). The results show these methods are ineffective for Process Reward Models, validating the necessity of our step-level causal approach.

**2. Methodological & Theoretical Clarifications**
*   **[R1, R3]Causal Grounding:** We clarified that our objective (Eq. 3) is derived directly from counterfactual causal decomposition, not heuristics. We empirically verified that simpler additive designs fail to match CoLD's performance.
*   **[R1]Implementation:** We provided detailed derivations for gradient propagation through the Pearson correlation and explained the noise injection mechanism.

**3. Interpretability, Robustness, and Efficiency**
*   **[R2]Interpretability:** We added visualization of attention maps and variance analysis to prove that the Bias Estimator specifically captures superficial length features rather than semantic reasoning.
*   **[R1, R2, R3, R4]Robustness & Cost:** We conducted sensitivity analyses demonstrating robustness to key hyperparameters ($\alpha$, $c$) and provided a runtime analysis showing minimal computational overhead.

We believe these revisions firmly establish the effectiveness, generality, and theoretical soundness of our approach.

---

### Meta-Review · Area_Chair_nBwX · 2026-01-07

**Summary:**

This paper identifies length bias in Process Reward Models (PRMs) and proposes CoLD (Counterfactually-Guided Length Debiasing), a framework combining length penalty, trainable bias estimator, and joint training. The authors demonstrate improvements on MATH500 and GSM-Plus benchmarks. However, the reception is predominantly negative even after the discussion period, with three of four reviewers recommending rejection due to missing critical baseline comparisons, disputed claims about "causality framework", limited evaluation scope, and reproducibility concerns.

**Reviewer Concerns:**

Some critical concerns were partially addressed during the discussion. In particular, the authors added additional baselines in response to reviewer feedback. They also challenged the methodology-related concern and provided further evaluations. However, the new results appear to require more thorough scrutiny, and the paper’s overall novelty still does not seem fully convincing to the reviewers.

**Reviewer Scores:**

The strongly negative reviewers remain negative, and only one weakly positive reviewer didn't further raise the score.

---

### Decision · Program_Chairs · 2026-01-26

Reject